# Text Descriptions are Compressive and Invariant Representations for Visual Learning

**Zhili Feng**                                                                          *zhilif@andrew.cmu.edu*
*Machine Learning Department*
*Carnegie Mellon University*

**Anna Bair**                                                                                  *abair@cmu.edu*
*Machine Learning Department*
*Carnegie Mellon University*

**J. Zico Kolter**                                                                          *zkolter@cs.cmu.edu*
*Computer Science Department*
*Carnegie Mellon University*
*Bosch Center for AI*

**Reviewed on OpenReview:** *https://openreview.net/forum?id=spo7O5Fyv0*

## Abstract

Modern image classification is based on directly predicting classes via large discriminative networks, which do not directly contain information about the intuitive visual features that may constitute a classification decision. Recently, work in vision-language models (VLM) such as CLIP has provided ways to specify natural language descriptions of image classes, but typically focuses on providing single descriptions for each class. In this work, we demonstrate that an alternative approach, in line with humans' understanding of multiple visual features per class, can also provide compelling performance in the robust few-shot learning setting. In particular, we introduce a novel method, *SLR-AVD (Sparse Logistic Regression using Augmented Visual Descriptors)*. This method first automatically generates multiple visual descriptions of each class via a large language model (LLM), then uses a VLM to translate these descriptions to a set of visual feature embeddings of each image, and finally uses sparse logistic regression to select a relevant subset of these features to classify each image. Core to our approach is the fact that, information-theoretically, these descriptive features are more invariant to domain shift than traditional image embeddings, even though the VLM training process is not explicitly designed for invariant representation learning. These invariant descriptive features also compose a better input compression scheme. When combined with finetuning, we show that SLR-AVD is able to outperform existing state-of-the-art finetuning approaches in both in-distribution and out-of-distribution tasks.

## 1 Introduction

Natural language supervised vision-language models (VLMs) like CLIP (Radford et al., 2021) create aligned image and text encoders via contrastive training. Unlike traditionally-trained classification networks, such alignment enables zero-shot image classification by prompting the text encoder with hand-crafted inputs like "`a photo of {}`" then predicting the target via the maximal inner product with the input image embedding. However, choosing effective prompts for zero-shot learning remains largely an ad-hoc process: Radford et al. (2021) has added several prompts like "`the cartoon {}`" or "`art of the {}`" aiming to improve ImageNet-R (Hendrycks et al., 2021a) performance, which (somewhat surprisingly) improved standard ImageNet accuracy as well. This has led to works that attempt to automatically extract relevant prompts from language models (Pratt et al., 2022), including work that uses these models to extract *multiple* visual

descriptors (Menon & Vondrick, 2022) then use the average prediction of these visual descriptions to classify the image.

In the few-shot setting, however, where a small amount of training data is available, a number of techniques can further improve classifier performance beyond zero-shot prompting alone. For example, it has become commonplace to finetune zero-shot classifiers via linear probing or other approaches (Kumar et al., 2022), including methods that interpolate between the zero-shot and finetuned classifiers (Wortsman et al., 2022) to achieve better out-of-distribution robustness. Alternatively, one can also adapt the prompts themselves using this few-shot data, using e.g. techniques from soft prompt tuning (Zhou et al., 2022b), though these learned prompts are not readable, nor are their nearest dictionary projections (Khashabi et al., 2021). Finally, recent work has also looked at ways to combine automatically-extracted prompts using few-shot learning (Yang et al., 2022), though this approach used a very specific learned weighting over such descriptions for interpretability purposes.

In this work, we investigate the visual learning problem with text descriptive features from an information-theoretic perspective. In particular, our motivation comes from two desiderata: compression and invariance (to domain shifts). The information bottleneck perspective encourages representations to compress the input as much as possible while maintaining high mutual information with the labels. On the other hand, the invariance principle favors representations that are less informative about the domains, in particular, the mutual information between the representations and the domain index should be small (Zhao et al., 2022; Li et al., 2021; 2022; Zhao et al., 2019; Arjovsky et al., 2019; Ahuja et al., 2021). Rooted in these information-theoretic principles, we propose a simple and effective method to generate classifiers based upon multiple automatically-extracted visual descriptors of each class. Our new method, *SLR-AVD (Sparse Logistic Regression using Augmented Visual Descriptors)*, uses a language model to extract multiple potential visual features of each class, then uses $\ell_1$-regularized logistic regression to fit a sparse linear classifier on top of these visual descriptions. The key observation that supports our method is that these descriptive features retain substantial information about the true labels, yet are not informative about the domain index, making them good invariant representations of the images. Additionally, these descriptive features are better input compressors and thus can generalize better.

Once the important visual descriptors are selected, we can also finetune the image encoder with the selected sparse pattern to further improve classification accuracies. Using this procedure, SLR-AVD outperforms baselines in both in-distribution (ID) and out-of-distribution (OOD) image classification across a range of image datasets. Specifically, SLR-AVD on ImageNet and its variations (including ImageNet-R, ImageNet V2, etc.) outperform linear probing with image features by 6.2% to 10.48% varying $k$-shot from $k = 1$ to $k = 32$. When combining SLR-AVD with WISE-FT (Wortsman et al., 2022), in the in-distribution task, our method outperforms standard finetuning by 1.43% with 1-shot, 1.62% with 2-shot, and 1.61% with 4-shot training data. When we average over five ImageNet variations, we outperform standard finetuning by 0.88% with 1-shot, 0.73% with 2-shot, and 0.64% with 4-shot training data.

**Notation** Throughout the paper, we use $g(\cdot)$ to denote the text encoder and $f(\cdot)$ to denote the image encoder. We use $\boldsymbol{t}$ for text tokens and $\boldsymbol{p}$ for images. For a vector $\boldsymbol{v}$, subscripted $\boldsymbol{v}_i$ represents the $i$th entry. We sometimes overload the notation $\boldsymbol{t}_c$ to represent a vector belonging to a class $c$, this should be clear from the context. We use $\mathcal{C}$ to denote the set of classes. We use $I(X; Y)$ to denote the mutual information between a pair of random variables $(X, Y)$.

## 2 Related works and motivation

### 2.1 Prompt tuning in VLMs

Contrastive VLMs aim to minimize the contrastive loss between matching image-text pairs. Let the image embedding be $f(\boldsymbol{p}) \in \mathbb{R}^{(1+M) \times d}$, the text embedding be $g(\boldsymbol{t}) \in \mathbb{R}^{(1+P) \times d}$, where $M, P \in \mathbb{R}$ denotes the number of tokens created by the transformer encoder, and $d \in \mathbb{R}$ is the dimension of each token's embedding. Without loss of generality, let the first entry of the embeddings be the [CLS] token, denote as $g(\boldsymbol{t})_0$. The

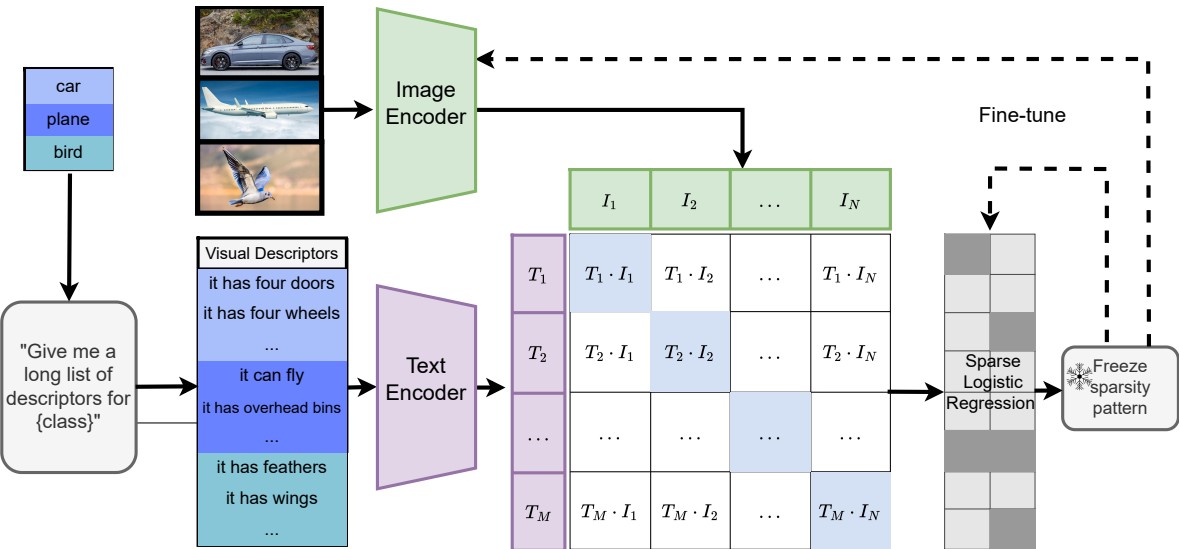

Figure 1: An overview of our proposed method. We prompt GPT-3 for a list of visual descriptors for each class and encode these texts. The image embeddings are instantiated to these descriptors by taking inner products. For an image embedding in $\mathbb{R}^d$, this operation projects it onto a $\mathbb{R}^M$ dimensional space, but it may live in a submanifold. We apply sparse logistic regression over all $\mathbb{R}^{n \times M}$ training data for feature selection. Finally, we **freeze** the sparsity pattern and finetune both the linear layer and the image encoder to align the image features with the visual descriptors.

probability of the prediction is then represented as: $p(y = c | \boldsymbol{p}, \boldsymbol{t}) = \frac{\exp\left(\langle f(\boldsymbol{p})_0, g(\boldsymbol{t}_c)_0 \rangle / \tau\right)}{\sum_{c'} \exp\left(\langle f(\boldsymbol{p})_0, g(\boldsymbol{t}_{c'})_0 \rangle / \tau\right)}$, where $\boldsymbol{t}_c$ is the zero-shot text prompt for class $c$. The class whose prompt has the largest inner product with the image embedding will be the zero-shot prediction. Zhou et al. (2022b) optimizes over the continuous text embedding space for the best prompts. Several follow-up works (Zhou et al., 2022a; Zhu et al., 2022) propose various prompt tuning methods for different task settings. The methods that eventually use $g(\boldsymbol{t}_c)_0$ are in essence regularized linear probing where the search space is constrained by the co-domain of $g(\cdot)_0$. Chen et al. (2022) uses local information of the image embedding $f_1, \ldots, f_{M+1}$ for optimizing an optimal transport distance between local image information and prompts. Lu et al. (2022) learns distributions over prompts for efficient adaptation to downstream recognition tasks. Wen et al. (2023) discusses discrete prompt search in the context of text-to-image settings.

Pratt et al. (2022) prompts LLMs for descriptions of each class and shows that these prompts can achieve better zero-shot image classification accuracy. Menon & Vondrick (2022) prompts LLMs to generate visual descriptors for image classification. For each class $c$, they query GPT-3 using the prompt "What are useful features for distinguishing a {c} in a photo?". A score is estimated for $c$ given an image $\boldsymbol{p}$: $s(c, \boldsymbol{p}) = \frac{1}{|D(c)|} \sum_{\boldsymbol{t} \in D(c)} \phi(\boldsymbol{t}, \boldsymbol{p})$, where $D(c)$ is the set of descriptors for $c$, and $\phi(\boldsymbol{t}, \boldsymbol{p}) = \langle f(\boldsymbol{p})_0, g(\boldsymbol{t})_0 \rangle$ is the inner product between the image and text embeddings. They show this average ensemble can outperform zero-shot classifiers while maintaining interpretability.

Similar to what we propose, LaBo (Yang et al., 2022) also considers per-class level descriptions in the few-shot setting. A key difference is that they perform a per-class level description filtering through submodular optimization, and they apply softmax to a linear weight $\sigma(\boldsymbol{W})$ to ensemble the selected features. On the other hand, we directly select features using sparse logistic regression. Our approach immediately gives both the important features and the coefficients and is statistically optimal under certain sparsity assumptions. One of the potential drawbacks of LaBo is their visual descriptions are filtered per-class level, which can hinder feature sharing between classes. LaBo uses $\sigma(\boldsymbol{W})$ in order to gain probabilistic interpretations of the features, while our emphasis on robustness only requires $\boldsymbol{W}$ to be sparse.

## 2.2 Robust fine-tuning of zero-shot models

There are numerous works that study robust finetuning of zero-shot models (Goyal et al., 2022; Kumar et al., 2022; Wortsman et al., 2022). In this work, we adopt the weight interpolation method WISE-FT to improve the OOD test accuracy (Wortsman et al., 2022). In general, let $\mathbf{\Phi}$ refer to any set of weights in the network (just the linear layer, linear layer + image encoder, etc). Let the finetuned weight be $\mathbf{\Phi}_{\text{learned}}$ and let the zero-shot predictor be $\mathbf{\Phi}_{\text{zs}}$. Wortsman et al. (2022) observes that while $\mathbf{\Phi}_{\text{learned}}$ performs better than $\mathbf{\Phi}_{\text{zs}}$ on ID tasks, it is worse at OOD tasks. Hence they propose to interpolate the two sets of weights as $\alpha\mathbf{\Phi}_{\text{learned}} + (1-\alpha)\mathbf{\Phi}_{\text{zs}}$. This surprisingly simple weight ensemble helps both in-distribution and out-of-distribution tasks. This method also naturally applies to linear probing by simply freezing the CLIP encoder throughout, and only training and interpolating the linear head.

## 2.3 Compression and Invariant Representation

The term "compression" has been given various meanings under different contexts. Arora et al. (2018) derived a PAC bound where generalization depends on the compression of the model parameters; Moran & Yehudayoff (2016) developed a sample compression scheme where both the features and labels are compressed; information bottleneck (Tishby & Zaslavsky, 2015) proposed to learn representations $Z$ that "compresses" the inputs $X$ by minimizing $I(X; Z)$ subject to some constraints. Blier & Ollivier (2018); Blum & Langford (2003) discussed label compression in terms of model description length. In this work, we use the term to represent input compression (as in the information bottleneck), such that the features contain little information about the inputs features. From a PAC-learning perspective, a better input compression will lead to a smaller generalization error (Shwartz-Ziv et al., 2018; Galloway et al., 2022), motivating our use of text descriptive features. A complementary idea from information theory is the invariance principle. The idea is that we want to learn representations that are very informative about the labels, but not so about the domain information. Mathematically, the principle encourages $\max_Z I(Y; Z) - \lambda I(Z; A)$ where $A$ is the domain information (Zhao et al., 2022). While it is understood that invariance by itself is insufficient for OOD generalization (Ahuja et al., 2021; Rosenfeld et al., 2020), algorithms based on the invariance principle still achieve competitive results on several OOD benchmarks (Koh et al., 2021).

## 3 Proposed method

In this section, we present our proposed method, SLR-AVD, summarized in fig. 1. We will discuss how to generate features, select a sparse set of useful descriptions, and finally, how to align the encoder in detail. We will also state how the proposed method aligns with information-theoretic principles.

## 3.1 Generating visual descriptors

To generate the visual descriptors for ImageNet and its variations, we first use the following prompt to query GPT-3: "`Give me a long list of descriptions for {}:`".

GPT-3 is quite sensitive to format instruction. Using the prompt "Give me a list" always leads to a list format, making it straightforward to select the useful text with regular expressions. Following the method in Menon & Vondrick (2022), we condition these descriptors on the class name, using texts of the form "$\{c\}$ `which has` $\{\boldsymbol{t}_c^i\}$" for each class $c$ and the $i$th descriptor. For each class $c$, we gather $M_c$ descriptors from GPT-3.

Furthermore, for each class, there exists a set of hand-crafted prompt templates like "`a photo of {}`" or "`an art of {}`". If there are $T$ total number of such templates, using the class name $c$, we can generate $T$ total prompt embeddings for each class. We take the average of these prompt embeddings *in addition to* the aforementioned visual descriptors, leading to $M_c + 1$ number of prompts for each class. For simplicity, we will refer to the GPT-3 generated text features as *visual descriptors (VD)*, the templates with class names as *class prompts* (CP), and the union as *augmented visual descriptors (AVD)*. We will also refer to their *embeddings* using the same names, which should be clear from the context.

Denote $M = \sum_{c \in \mathcal{C}} M_c$ where $\mathcal{C}$ is the set of all classes. The visual descriptors, class prompts, and augmented visual descriptors can be encoded into three matrices $\boldsymbol{U}_{\text{vd}} \in \mathbb{R}^{M \times d}, \boldsymbol{U}_{\text{cp}} \in \mathbb{R}^{|\mathcal{C}| \times d}, \boldsymbol{U}_{\text{avd}} \in \mathbb{R}^{(M+|\mathcal{C}|) \times d}$. Given an image embedding $\boldsymbol{z} := f(\boldsymbol{p})_0 \in \mathbb{R}^d$, these three matrices respectively created three sets of new features $\boldsymbol{h}_{\text{vd}} = \boldsymbol{U}_{\text{vd}}\boldsymbol{z}$, $\boldsymbol{h}_{\text{cp}} = \boldsymbol{U}_{\text{cp}}\boldsymbol{z}$, and $\boldsymbol{h}_{\text{avd}} = \boldsymbol{U}_{\text{avd}}\boldsymbol{z}$. Notice that all three $\boldsymbol{U}$ matrices are fixed and never trained. We call the action of inner product $\langle \boldsymbol{U}, \cdot \rangle$ as "instantiating". We will also refer to the instantiated features $\boldsymbol{h}$ as the (text/language) descriptive features. Given $\boldsymbol{h}$, we can learn three matrices $\boldsymbol{W}_{\text{vd}} \in \mathbb{R}^{|\mathcal{C}| \times M}, \boldsymbol{W}_{\text{cp}} \in \mathbb{R}^{|\mathcal{C}| \times |\mathcal{C}|}, \boldsymbol{W}_{\text{avd}} \in \mathbb{R}^{|\mathcal{C}| \times (M+|\mathcal{C}|)}$.

Setting $\boldsymbol{W}_{\text{vd}} = \text{blkdiag}\left(\left(\underbrace{\frac{1}{|M_c|}, \ldots, \frac{1}{|M_c|}}_{|M_c| \text{ copies}}\right)_{c \in \mathcal{C}}\right)$, then $\boldsymbol{W}_{\text{vd}}\boldsymbol{U}_{\text{vd}}$ leads to the average ensemble in Menon & Vondrick (2022). Here $\text{blkdiag}(A_1, \ldots, A_n)$ creates the block diagonal matrix:

$$\text{blkdiag}(A_1, \ldots, A_n) = \begin{bmatrix} A_1 & 0 & \cdots & 0 \\ 0 & A_2 & \cdots & 0 \\ \vdots & \vdots & \ddots & 0 \\ 0 & 0 & \cdots & A_n \end{bmatrix}.$$

Setting $\boldsymbol{W}_{\text{cp}} = I_{|\mathcal{C}| \times |\mathcal{C}|}$, we get back the zero-shot classifier $\boldsymbol{W}_{\text{zs}} = \boldsymbol{W}_{\text{cp}}\boldsymbol{U}_{\text{cp}}$. One can naturally merge $\boldsymbol{W}_{\text{vd}}$ and $\boldsymbol{W}_{\text{cp}}$ into $\boldsymbol{W}_{\text{avd}} = [\boldsymbol{W}_{\text{vd}}, \boldsymbol{W}_{\text{cp}}]$, which we use in our proposed method. We note that these three $\boldsymbol{W}$ matrices can all serve as zero-shot classifiers. During inference, the prediction is made by picking $\arg\max_{i \in [\mathcal{C}]} (\boldsymbol{W}\boldsymbol{U}\boldsymbol{z})_i$.

## 3.2 Learning sparse ensemble and aligning the image encoder

The previously defined matrix $\boldsymbol{U}_{\text{avd}}$ can be viewed as a linear projection of the image embedding onto a $M+|\mathcal{C}|$ dimensional semantic space. While this space has a high ambient dimension, the projected embeddings live in a low-dimensional manifold that has rank less than or equal to that of the image embedding space. By enforcing a sparsity constraint on $\boldsymbol{W}_{\text{avd}}$, we can select the most important dimensions among $\boldsymbol{h}_{\text{avd}}$. We demonstrate that the selected subspace is also robust to natural distribution shifts. Intuitively, we imagine that the large distribution shift in the image embedding space only corresponds to a small shift in the semantic space, since the semantics of images should be invariant. We will later demonstrate with mutual information estimations.

With a fixed $\boldsymbol{U}_{\text{avd}}$, we learn $\boldsymbol{W}_{\text{avd}}$ with $\ell_1$ regularization $\|\boldsymbol{W}_{\text{avd}}\|_1$ and the cross-entropy loss. Not only does sparse logistic regression select the important features, but it actually also finds the intuitive features. For example, on CIFAR-10, we demonstrate that the selected features are usually the ones that actually describe that class: for each class, we pick the three features with the largest coefficients, and show that the properly descriptive class features are chosen most often; the results are listed in table 9 in the appendix. After obtaining a sparse $\widehat{\boldsymbol{W}}_{\text{avd}}$, we fix $\boldsymbol{U}_{\text{avd}}$ and the *sparsity pattern* of $\widehat{\boldsymbol{W}}_{\text{avd}}$, and finetune both the image encoder $f$, as well as the entries in $\widehat{\boldsymbol{W}}_{\text{avd}}$. This process aligns with LP-FT (Kumar et al., 2022), which has some theoretical justification for its robustness.

## 3.3 Text descriptive features are compressive and invariant

Beyond the improvement in performance alone, however, the core of our method relies on the empirical evidence that text descriptive features have many benefits from an information-theoretic perspective. Specifically, we show here that the text descriptive features form more *invariant* and more *compressive* representations of the data than the naive image encoder features. This motivates their use, especially under distribution shift, where we see them outperform the alternatives.

We base our investigation upon two notions: the invariance principle and the information bottleneck. First, the invariance principle from causality (Pearl, 1995) states that the predictors should only rely on the causes of the labels rather than the spurious features. Following this principle, several mutual information (MI) based OOD generalization works (Arjovsky et al., 2019; Zhao et al., 2022; Li et al., 2021; 2022; Zhao et al.,

2019; Feng et al., 2021; Ahuja et al., 2021) propose that a good feature representation $Z$ would have high mutual information with the label, $I(Z;Y)$, but low MI with the domain index, $I(Z;A)$, so as not to leak information about the domain itself. Closely related is the information bottleneck, which similarly states that a good representation will again have high MI with the label, but low MI with the input $I(Z;X)$. In recent years, several works have suggested that combining the invariance principle with the information bottleneck can lead to practical and provably strong OOD generalization (Ahuja et al., 2021; Li et al., 2022).

We demonstrate that the text descriptive features essentially obey both the tenets of the invariance principle and the information bottleneck: the extracted text features $H$ have high MI with the labels, but substantially lower MI with both the domain index and the input itself. The features of our framework correspond to the following Markov chain:

$$Y \rightarrow X \xrightarrow{f(\cdot)_0} Z \xrightarrow{\boldsymbol{U}} H \xrightarrow{\boldsymbol{W}} \hat{Y}, \tag{1}$$

where $\boldsymbol{y} \sim Y, \boldsymbol{p} \sim X, \boldsymbol{z} \sim Z, \boldsymbol{h} \sim H, \hat{\boldsymbol{y}} \sim \hat{Y}$ corresponds to realizations of the true labels, the input images, the image embeddings, the text descriptive features, and the predictions (the capital letters are random variables) respectively. Here $\boldsymbol{W}, \boldsymbol{U}, \boldsymbol{h}$ and $H$ can be subscribed by avd, vd, cp as in section 3. We will use $A$ for the domain index.

By the Data Processing Inequality (DPI, Cover (1999)), we immediately have that $I(X;Y) \geq I(Z;Y) \geq I(H;Y)$. Additionally, however, we also observe for the text descriptive features $I(H;Y)$ is nearly as large as $I(Z;Y)$ (i.e., there is not much decrease in the information about the label), but $I(H;A)$ and $I(H;X)$ are substantially lower than $I(Z;A)$ and $I(Z;X)$ (i.e, the text descriptive features leak much less information about the domain index and the input).

To assess this, we conduct numerical evaluations on CIFAR-10 (Krizhevsky et al., 2009), CIFAR-10.1 (Recht et al., 2018), and CIFAR-10.2 (Lu et al., 2020). We index these three datasets, denoting the index random variable as $A$. We compute the image embedding $\boldsymbol{z}$ and the instantiated descriptive feature $\boldsymbol{h}$ for every image in these three test sets. To estimate mutual information, we use the SMILE estimator (Song & Ermon, 2019). The numerical estimation is presented in fig. 2. MI is estimated for two sets of text descriptive features: $\boldsymbol{h}_{\text{cp}} \sim H_{\text{cp}}$ and $\boldsymbol{h}_{\text{avd}} \sim H_{\text{avd}}$. Importantly, $H_{\text{cp}}$ should be viewed as a post-processing of $H_{\text{avd}}$. Intuitively, we see that $I(Z;Y) > I(H_{\text{avd}};Y) > I(H_{\text{cp}};Y)$ by DPI. We also see that $I(Z;A) > I(H_{\text{avd}};A) > I(H_{\text{cp}};A)$, which suggests that the text descriptive features $\boldsymbol{h}$ are much more invariant to the distribution shift. The noticeable gap between $I(H_{\text{avd}}, Y)$ and $I(H_{\text{cp}}, Y)$ explains why it is beneficial to work with text descriptive features beyond vanilla zero-shot classification.

From the information bottleneck perspective, Figure 2 also presents that $I(X;H_{\text{avd}}) < I(X;Z)$ by a large margin, we can then interpret $H_{\text{avd}}$ as a "better" compression of the input image $X$, in the sense that it preserves only information in $X$ that is helpful for predicting $Y$. Of course, this also means that one cannot reconstruct $X$ from $H_{\text{avd}}$ better than from $Z$, although this is an orthogonal goal to ours. Typically better input compressions lead to smaller generalization error. Under mild conditions one can bound the generalization error of feature $Z$ with probability at least $1 - \delta$: $\text{GenErr} \leq \sqrt{\frac{2I(X;Z) + \log(2/\delta)}{n}}$, where $n$ is the number of training samples (Shwartz-Ziv et al., 2018). Intuitively, if the features have small MI with the inputs, then the perturbation in the input space cannot perturb the features too much, hence constraining the expressiveness of the features. Since $I(H_{\text{avd}};X)$ is significantly smaller than $I(Z;X)$, we can expect a more predictable test performance (compared to the training performance). On the other hand, high $I(H_{\text{avd}};Y)$ makes sure that the accuracy will not be too low. The synergy of the two notions elucidates the superiority of AVD in the few-shot setting.

In addition, we also measure $I(\widehat{H}_{avd};X)$, $I(\widehat{H}_{avd};Y)$, and $I(\widehat{H}_{avd};A)$, where $\widehat{H}_{avd}$ represent the AVD selected by the $\ell_1$ regularizer. By DPI, we immediately have $I(H_{avd};Y) \geq I(\widehat{H}_{avd};Y)$, in fig. 2, we see that numerical evaluation shows nearly identical estimation for these two statistics, indicating that most information about the label is preserved by the sparsity regularization. Meanwhile by DPI, we also immediately have $I(H_{avd};A) \geq I(\widehat{H}_{avd};A)$, and both are negligibly small. The interesting part comes in $I(H_{avd};X) \geq I(\widehat{H}_{avd};X)$. Once again, DPI tells us the inequality should hold true, and numerical evaluation suggests that $\widehat{H}_{avd}$ has a noticeable drop in its mutual information with $X$, explaining why they generalize better. We remark that in measuring mutual information with variational methods, there are inevitable

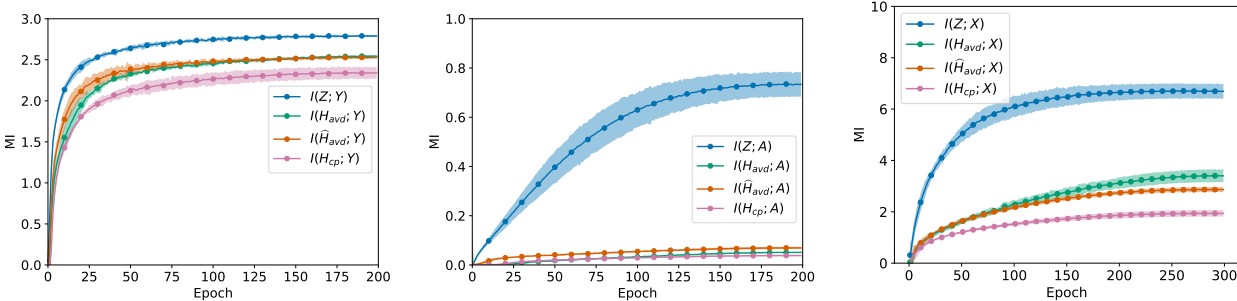

Figure 2: The MI estimations at interest. The estimator is variational and we include the whole optimization trajectory to show convergence (hence we can focus on the estimations in the last epoch). Each experiment is run three times and two standard deviations are filled. **Left:** the MI between a different set of features and the labels. **Middle:** the MI between a different set of features and the domain indices. **Right**: the MI between a different set of features and the input images.

Table 1: Accuracies on ImageNet (IN) and its variants. We compare LP vs SLR-AVD.

| Shots | $k = 1$ | | $k = 2$ | | $k = 4$ | | $k = 8$ | | $k = 16$ | | $k = 32$ | |
|---|---|---|---|---|---|---|---|---|---|---|---|---|
| Methods | LP | SLR-AVD | LP | SLR-AVD | LP | SLR-AVD | LP | SLR-AVD | LP | SLR-AVD | LP | SLR-AVD |
| IN | 31.51 | **40.56** | 44.06 | **54.16** | 54.66 | **63.19** | 62.33 | **68.23** | 67.55 | **71.40** | 71.15 | **73.67** |
| IN-R | 35.23 | **48.88** | 46.30 | **61.23** | 54.50 | **67.64** | 59.25 | **70.58** | 62.16 | **72.54** | 64.32 | **74.53** |
| IN-A | 22.52 | **29.81** | 27.26 | **36.96** | 32.34 | **42.09** | 34.88 | **44.41** | 36.68 | **45.15** | 39.19 | **47.89** |
| IN-V2 | 26.91 | **35.12** | 37.13 | **47.07** | 45.92 | **55.02** | 52.50 | **59.15** | 57.62 | **62.52** | 61.23 | **64.75** |
| IN-Sketch | 16.80 | **22.87** | 21.96 | **31.03** | 28.77 | **37.43** | 33.29 | **40.73** | 35.62 | **42.94** | 38.64 | **45.39** |
| ObjectNet | 19.38 | **25.43** | 24.98 | **34.11** | 32.44 | **40.39** | 36.02 | **42.80** | 41.50 | **45.82** | 43.67 | **49.17** |
| Average ↑ | | 8.39 | | 10.48 | | 9.52 | | 7.94 | | 6.54 | | 6.20 |

estimation and optimization errors – for example, even though DPI tells us $I(H_{avd}; A) \geq I(\widehat{H}_{avd}; A)$, our measurement leads to its contrary, but both values are vanishingly small and hence the difference is insignificant. On the other hand, our focus here is the significant difference between $H_{avd}$ (as well as $\widehat{H}_{avd}$) and $Z$, that is, AVDs are much more invariant and compressible compared to their image embedding counterparts.

## 4  Experiment

We demonstrate the improved performance of AVD in three settings: zero-shot (no parameter update), linear probing (only the last layer is updated), and full model finetuning. Throughout the experiments, we focus on the few-shot setting. We test our method on ImageNet, ImageNet-R, ImageNet-V2, ImageNet-A, ImageNet-Sketch, and ObjectNet (Deng et al., 2009; Hendrycks et al., 2021a;b; Recht et al., 2019; Wang et al., 2019; Barbu et al., 2019), demonstrating the superiority of the sparsely learned visual descriptors ensemble. We abbreviate ImageNet as IN, and correspondingly their variation, for example, we write ObjectNet as IN-Object. By default, we use the ViT-B/16 model unless otherwise specified. The hand-crafted templates for ImageNet classes contain a set of seven prompts suggested in the CLIP github repository (https://github.com/openai/CLIP): 1. "`itap of a {}.`" 2. "`a bad photo of the {}.`" 3. "`a origami {}.`" 4. "`a photo of the large {}.`' 5. "`a {} in a video game.`" 6. "`art of the {}.`" 7. "`a photo of the small {}.`" This set usually outperforms the original 80 templates in Radford et al. (2021).

For simplicity, we will use the following acronyms for different methods and datasets, also see table 2 for a detailed comparison among each methods. Whenever we combine our method with WISE-FT (Wortsman et al., 2022), it will be explicitly mentioned. We defer the hyperparameter discussions to the appendix.

**ZS:** Zero-shot classification using text embeddings of hand-crafted prompts ensembles.

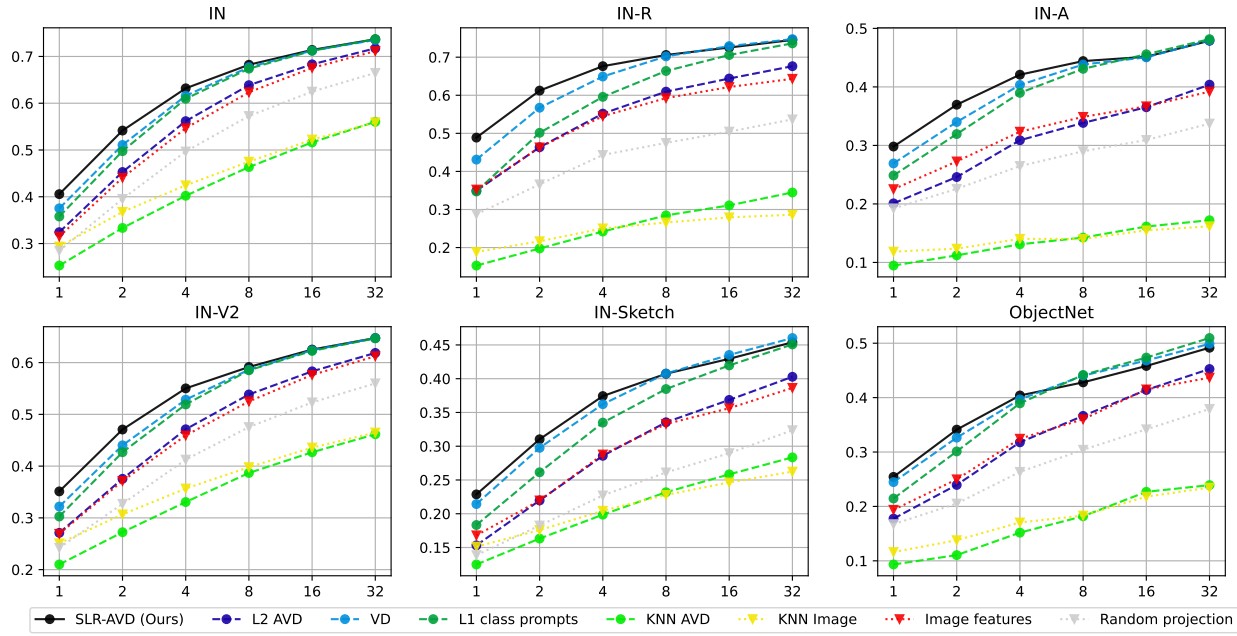

Figure 3: Few-shot experiments compare several baseline methods vs SLR-AVD. In each subfigure, the x-axis represents the number of shots per class, the y-axis represents test accuracy on ImageNet (IN) variants and ObjectNet. Here we consider shot in $\{1, 2, 4, 8, 16, 32\}$ shots per class. SLR-AVD is more sample efficient in the in-distribution reference and is also much more robust to several distribution shifts.

Table 2: Acronyms for several methods in consideration. In the column heads, **CP**: class prompts; **VD**: visual descriptors; **Img**: image embeddings.

|  | Features | | | Parameter updates | |
|---|---|---|---|---|---|
|  | CP | VD | Img | Linear | All |
| ZS | ✓ |  |  |  |  |
| ZS-VD |  | ✓ |  |  |  |
| ZS-AVD | ✓ | ✓ |  |  |  |
| LP |  |  | ✓ | ✓ |  |
| SLR-AVD | ✓ | ✓ |  | ✓ |  |
| FT |  |  | ✓ | ✓ | ✓ |
| SLR-FT-AVD | ✓ | ✓ |  | ✓ | ✓ |

**ZS-VD, ZS-AVD:** Zero-shot classification using visual descriptor and augmented visual descriptors, respectively.

**LP:** Linear probing using image embeddings.

**SLR-AVD:** Sparse logistic regression using AVDs.

**FT:** Finetuning the image encoder and classification head.

**SLR-FT-AVD:** Sparse logistic regression with AVD, and then finetune the linear head plus the image encoder with frozen sparsity patterns.

Table 3: Accuracies of zero-shot, visual descriptors, and augmented visual descriptors on ImageNet (IN) and its variants. ZS-AVD uses descriptors from GPT4 and ZS-AVD$_2$ uses descriptors from Llama2-13B-chat. ZS-AVD outperforms all baselines across different datasets.

| | ZS | ZS-VD | ZS-AVD | ZS-AVD$_2$ | Waffle-2 | Waffle-5 | Waffle-10 |
|---|---|---|---|---|---|---|---|
| IN | 68.78 | 65.89 | **69.52** | 69.15 | 64.36 | 62.27 | 60.24 |
| IN-V2 | 62.23 | 59.19 | **62.97** | 62.86 | 57.95 | 56.39 | 54.54 |
| IN-R | 77.72 | 72.75 | 77.85 | **77.88** | 73.32 | 72.83 | 70.68 |
| IN-A | 50.64 | 46.11 | 50.87 | **51.07** | 46.4 | 44.51 | 50.65 |
| IN-Sketch | 48.38 | 44.84 | **48.91** | 48.45 | 44.43 | 42.90 | 41.57 |
| ObjectNet | 54.31 | 49.60 | **54.58** | 53.57 | 48.73 | 48.37 | 46.34 |

Table 4: WISE-FT weight interpolation for standard zero-shot/finetuning (FT) and SLR-FT-AVD with optimal $\alpha$. Accuracies are on ImageNet (IN) and its variations.

| Shot | $k = 1$ | | $k = 2$ | | $k = 4$ | |
|---|---|---|---|---|---|---|
| Method | FT | SLR-FT-AVD | FT | SLR-FT-AVD | FT | SLR-FT-AVD |
| IN | 68.88 | **70.31** | 69.59 | **71.21** | 70.48 | **72.09** |
| Average ↑ | | 1.43 | | 1.62 | | 1.61 |
| IN-R | 77.82 | **78.29** | 78.13 | **78.53** | 78.32 | **78.59** |
| IN-A | 50.09 | **51.29** | 50.43 | **51.51** | 52.11 | **52.64** |
| IN-V2 | 62.32 | **63.74** | 63.07 | **64.37** | 63.50 | **65.30** |
| IN-Sketch | 48.45 | **49.35** | 48.75 | **49.63** | 48.99 | **49.92** |
| ObjectNet | 54.52 | **54.94** | **55.01** | 54.99 | **55.77** | 55.41 |
| Average ↑ | | 0.88 | | 0.73 | | 0.64 |

## 4.1 Zero-shot with AVDs

As mentioned in section 3.1, we can easily establish zero-shot matrices with AVDs. We set $\boldsymbol{W}_{\text{vd}}$ to be the aforementioned block diagonal form, and $\boldsymbol{W}_{\text{cp}}$ to be an identity matrix. We merge them into $\boldsymbol{W}_{\text{avd}} = [\boldsymbol{W}_{\text{vd}}, \gamma \boldsymbol{W}_{\text{cp}}]$. Their performances are compared in table 7. ZS-AVD outperforms every zero-shot baseline on all ImageNet variations. We find that simply using VD usually underperforms ZS, indicating that the class names are one of the strongest prompts. This observation is intuitive as during contrastive training the class name itself is likely to show up in the caption the most often, compared to other visual descriptors. One can certainly try to improve ZS-VD results by more carefully prompting GPT-3, or gathering descriptors from different data sources/search engines. Pratt et al. (2022); Yang et al. (2022); Menon & Vondrick (2022) have studied the quality of descriptors across different datasets and hyperparameters (e.g. temperature for sampling, etc) settings. Here, we do not further pursue this direction. Instead, we utilize our observation that simply using the merged prompts $\boldsymbol{W}_{\text{avd}}$ already surpasses the best zero-shot classifier. Notice here we have a parameter $\gamma$ that decides how much we weight the zero-shot model. Empirically we find that setting $\gamma = 5$ is sufficient for all datasets. We conduct small-scale experiments on CIFAR-10 and its variations to further investigate the influence of difference choice of $\gamma$, the GPT prompts, and the GPT sampling hyperparameters. We find these choices typically do not lead to significant deviations in test accuracies unless the generated visual descriptors are too repetitive, see the appendix for details.

Another interesting baseline zero-shot method we consider is WaffleCLIP (Roth et al., 2023), where the prompt contains the ground truth class name and $p$ random tokens. Here, we compare to $p = 2, 5, 10$, and we notice that these random prompts are less informative compared to AVD.

## 4.2 Comparison to linear probing

We compare SLR-AVD to LP with $\{1, 2, 4, 8, 16, 32\}$ shots per class. Each experiment is conducted three times with independent random seeds. We report the averaged test accuracy on ImageNet and its distribution shift variations, see fig. 3 for details. Our proposed method outperforms linear probing on all tasks. Detailed accuracies are presented in table 1. In a nutshell, our method outperforms linear probing by 8.39%, 10.48%, 9.52%, 7.94%, 6.54%, 6.20% on $k = 1, 2, 4, 8, 16, 32$ respectively. When training with large shot size ($k = 32, 64, 256, 1024$), SLR-AVD still shows effectiveness, see fig. 16 in the appendix.

In addition to linear probing, we additionally consider several baselines in fig. 3: (1) AVD with $\ell_2$ regularization. (2) $k$NN classifier ($k = 3$) with image embeddings. (3) $k$NN classifier ($k = 3$) with AVD. (4) Randomly project the image embeddings onto $\mathbb{R}^{200}$ and perform linear probing. (5) VD. (6) Class prompts with $\ell_1$ regularizer.

Our proposed SLR-AVD outperforms all baselines. Specifically, we see that the incorporation of $\ell_1$ sparsity is crucial for the improvement of the few-shot performance; at the same time, the combination of both class prompts and VD is also important. The comparison to random projection also reveals that the benefit of sparse AVD is not simply due to its smaller dimension: the sparsely chosen AVDs are semantically meaningful, which comes with a stronger regularization effect.

Although learning with visual descriptors significantly outperforms linear probing in the few-shot setting, we should remark that ImageNet and its variations are usually considered "in-distribution" to the CLIP training data. In this case, the zero-shot model itself is usually a very strong baseline, and typically outperforms few-shot models, as can be observed by comparing the results in table 7 and table 1. WISE-FT serves as a strong method to improve both in-distribution and out-of-distribution accuracies. We can apply WISE-FT to any of our existing settings, including SLR-AVD and LP. In particular, we can train a linear head (and/or image encoder, depending on the setting) $\boldsymbol{W}_{\text{learned}}$, and interpolate with the zero-shot weight $\boldsymbol{W}_{\text{zs}}$ by taking a convex combination $\alpha \boldsymbol{W}_{\text{zs}} + (1 - \alpha) \boldsymbol{W}_{\text{learned}}$, for $\alpha \in \{\alpha_1, \ldots, \alpha_n\}$. We are free to vary $\alpha$. Then for each $\alpha_i$, we can plot that weight ensemble's ID and OOD test accuracy. This procedure thus creates an ID-OOD frontier and along the curve, some ensemble excels at both ID and OOD distribution. In the ID-OOD curves in fig. 4, we show the plot of $k = 4, 8, 16$. SLR-AVD's ID-OOD curve overwhelms that of LP, indicating that SLR-AVD is better at both ID and OOD tasks.

## 4.3 Comparison to finetuning

We compare WISE-FT where we additionally interpolate the image encoder, to a weight interpolation between SLR-FT-AVD and ZS-AVD (WISE-FT+SLR-AVD). The ID-OOD frontier is presented in fig. 4 and the accuracies are reported in table 4.

On the ID task, WISE-FT+SLR-FT-AVD outperforms vanilla WISE-FT by 1.43%, 1.62%, and 1.61% respectively with $k = 1, 2, 4$ shot training data. Averaging over 5 distribution shift datasets, with optimal $\alpha$, WISE-FT+SLR-FT-AVD outperforms vanilla WISE-FT by 0.88%, 0.73%, and 0.64% respectively for $k = 1, 2, 4$. The optimal $\alpha$ is picked independently for each method on each dataset.

## 4.4 Comparison to CoOp

CoOp and CoCoOp both tackle the few-shot learning problem from a prompt optimization perspective (Zhou et al., 2022b;a). Since the latter requires an additional neural adapter module and its benefit mostly lies in generalization to unseen classes, we only compare to CoOp in this work as it is more relevant to our setting. CoOp learns the prefix of "`[prefix] {classname}`" in the continuous token embedding space. The benefit of CoOp is that it operates in a continuous space, hence one can optimize using standard backpropagation. On the other hand, due to the requirement of backprop, CoOp stores a large computation graph, hence memory-efficiency is a big advantage of SLR-AVD over CoOp.

When implementing CoOp, we choose a prefix of length 16 and do not use a suffix. The prefix is fixed for all classes. We train with Adam for 20 epochs, setting the batch size to 512. This gives us comparable results to those of the original paper.

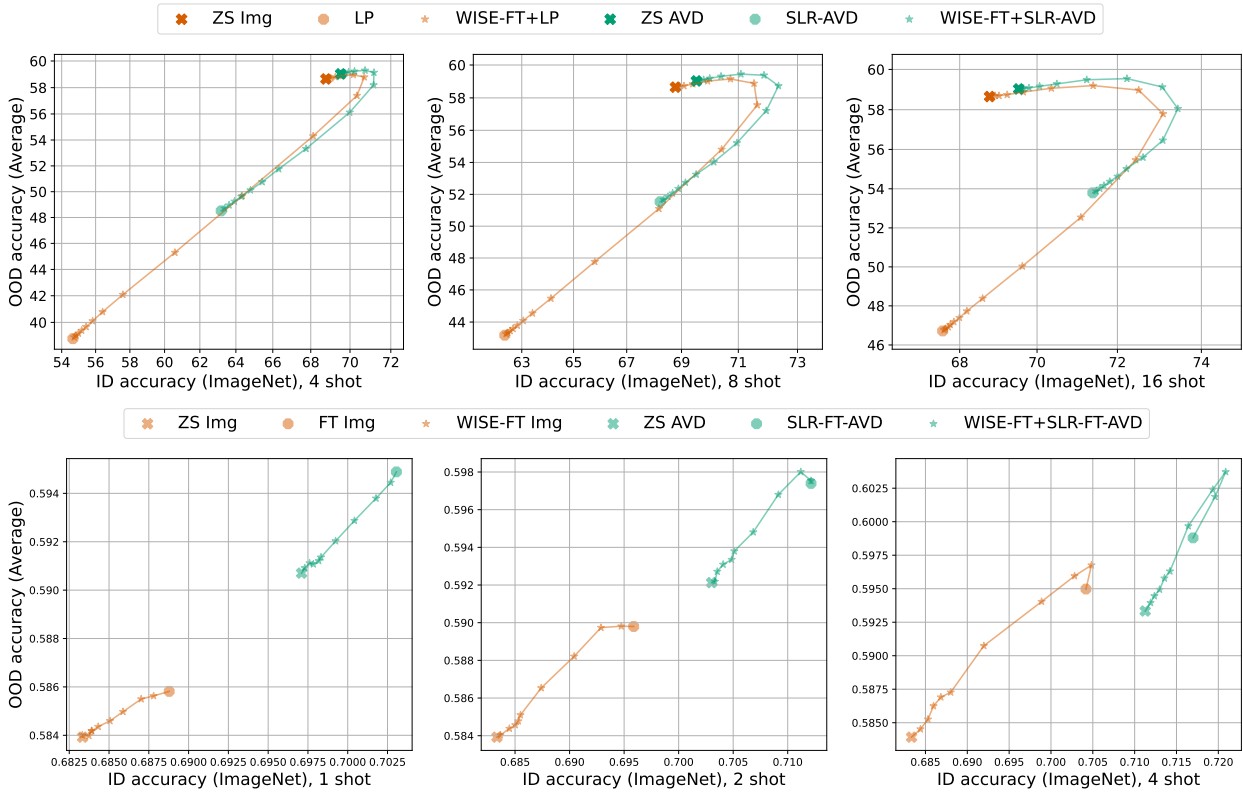

Figure 4: **Top:** ID-OOD accuracy curve of WISE-FT+LP vs WISE-FT+SLR-AVD. ID is tested on ImageNet, and OOD is averaged over 5 ImageNet variations. Experiments with $[4, 8, 16]$-shots are presented. Each accuracy is averaged over 3 runs. We can see that our proposed method overwhelms LP in all cases. **Bottom:** ID-OOD accuracy curve of WISE-FT vs WISE-FT+SLR-FT-AVD, full parameter updates are performed. ID is tested on ImageNet, and OOD is averaged over 5 ImageNet variations. Experiments with $[1, 2, 4]$-shots are presented. Each accuracy is averaged over 3 runs. We can see that our proposed method overwhelms WISE-FT in all cases.

Table 5: Accuracies of CoOp and SLR-AVD on ImageNet. Both methods are incorporated with WISE-FT. The results are reported with the best interpolation.

| Shots | 1 | 2 | 4 | 8 | 16 | 32 |
|---|---|---|---|---|---|---|
| CoOp | 69.54 | 69.73 | 70.14 | 70.55 | 71.11 | 71.85 |
| SLR-AVD | **69.83** | **70.33** | **71.18** | **72.37** | **73.45** | **74.34** |
| $\Delta$ | +0.28 | + 0.61 | + 1.04 | + 1.82 | + 2.33 | +2.49 |

For a fair comparison, we compare WISE-FT+CoOp to WISE-FT+SLR-AVD. The vision backbone used is ViT-B/16 for both methods. We use the ZS weight for CoOp WISE-FT interpolation. The results are reported in table 5, and we pick the interpolation that yields the best test accuracy. Our proposed method outperforms CoOp in all cases.

## 4.5 Comparison to LaBo

LaBo (Yang et al., 2022) proposes a similar approach to ours, where they first achieve many descriptors from various sources, then perform a feature selection based on submodular optimization, and finally they learn a classification layer using the selected features. There are two inconveniences that come with a submodular optimization based method: one is the users have to design a proper submodular function, and the other is

the feature selection and learning process are separated, imposing additional effort. For fair comparison, we compare to LaBo using CLIP ViT-L/14 backbone on ImageNet. We take the same set of concepts collected in Yang et al. (2022), randomly selected six features for each class, and train SLR-AVD using the subsampled features. While the original LaBo paper select 50000 features for 1000 classes, SLR-AVD selects 5961, 5890, 6678, 6959, and 6990 out of the total 7000 features, for $k = 1, 2, 4, 8, 16$, respectively. Notice that the actual matrix $\widehat{\boldsymbol{W}}_{\text{avd}} \in \mathbb{R}^{1000 \times 7000}$ is very sparse – roughly 98% to 99% percent of the entries are zero. The results are presented in table 6. SLR-AVD has outperformed both baselines.

Table 6: A comparison among LP, LaBo, and SLR-AVD. The accuracies of LP and LaBo are gathered from Yang et al. (2022), the image features are extracted from CLIP-L/14 for all models. This evaluation is done on ImageNet.

| Shots | 1 | 2 | 4 | 8 | 16 |
|---|---|---|---|---|---|
| LP | 42.25 | 55.71 | 64.80 | 71.23 | 75.08 |
| LaBo | 51.09 | 57.43 | 62.94 | 68.45 | 72.60 |
| SLR-AVD | **63.09** | **71.26** | **75.46** | **77.29** | **78.88** |

## 4.6 WILDS Benchmark

We further conduct experiments on the WILDS benchmark (Koh et al., 2021), specifically, iWildCam (Beery et al., 2021) and FMoW (Christie et al., 2018). Following the norm, on iWildCam, we evaluate based on the macro F1 for both ID and OOD test data; on FMoW, we evaluate based on accuracy for ID test data, and "worst geographical region" accuracy on OOD test data, see Koh et al. (2021) for details. In this section, we train with the full dataset instead of the few-shot data. The results are presented in table 7. We see that AVD does not always improve the performance in the zero-shot case, there is an improvement in iWildCam but a decrease in FMoW. On the other hand, SLR-AVD outperforms LP on iWildCam significantly, and perform on par with LP on FMoW. Meanwhile, learning with GPT-4 generated AVD does not have a strong edge over learning with those generated by a less capable language model, Llama2-13B-chat (Touvron et al., 2023). A very compelling reason is that contrastive multi-modal objective function ignores subtle semantic differences, for example, CLIP treats texts like a bag of words (Yuksekgonul et al., 2022). We leave further investigation to future research.

## 4.7 Ablation study

In this section, we try to understand what makes SLR-AVD outperform other baselines. In particular, we would like to investigate two factors: the importance of sparse regularization and the language models that generate the AVDs. We conduct these results using the iWildCam dataset, the results are presented in table 8. We see that while $\ell_2$ regularization does improve the performance on the ID data, it is not necessarily more robust to the OOD dataset, while $\ell_1$ improves both ID and OOD performance. On the other hand, a less capable LLM can still generate reasonable AVD that match the performance of AVD generated by GPT-4.

Table 7: Evaluation on WILDS benchmark. ZS-AVD$_1$ and SLR-AVD$_1$ use descriptors from GPT-4, ZS-AVD$_2$ and SLR-AVD$_2$ use descriptors from Llama2-13B-chat.

| | ZS | ZS-AVD$_1$ | ZS-AVD$_2$ | LP | SLR-AVD$_1$ | SLR-AVD$_2$ |
|---|---|---|---|---|---|---|
| iWildCam ID | 10.84 | 13.12 | 11.87 | 40.20 | 43.06 | **43.61** |
| iWildCam OOD | 8.82 | 11.32 | 11.04 | 28.88 | **31.91** | 31.09 |
| FMoW ID | 20.45 | 20.31 | 21.28 | 46.87 | **47.21** | 46.63 |
| FMoW OOD | 18.92 | 17.79 | 19.34 | 29.00 | 28.89 | **30.00** |

Table 8: Ablation studies on the iWildCam dataset. SLR-AVD$_1$ is trained with GPT-4 generated AVD, and SLR-AVD$_2$ is with Llama2-13B-chat generated AVD.

|  | LP | $\ell_2$-AVD | SLR-AVD$_1$ | SLR-AVD$_2$ |
|---|---|---|---|---|
| ID | 40.20 | 42.95 | 43.06 | **43.61** |
| OOD | 28.88 | 26.93 | **31.91** | 31.09 |

## 5  Conclusion

Motivated by the invariance principle and information bottleneck, we present how to leverage descriptive features for image learning in the few-shot setting robustly. These descriptive features can be easily obtained from LLMs. Applying sparse logistic regression then successfully selects the important features, which turn out to be intuitive. Our proposed method outperforms linear probing and standard finetuning in both ID and OOD tasks, with or without combining with WISE-FT. This approach helps us further understand the CLIP embedding space and how the semantics serve as a strong robust prior. Moving forward, it is important to understand and quantify the robustness of the visual descriptors' space and compare it to the image embedding space statistically. On the practical side, this work aligns image encoders to a fixed text encoder; it is valuable to study how to simultaneously align both encoders in a robust way.

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

## A   Miscellaneous

**Hyperparameter**   For ImageNet and its variations, we fix a set of 6804 augmented visual descriptors. The hyperparameters are swept over disjoint training and validation sets of size 20 per class for LP and SLR-AVD. For $\ell_1$ regularization, its non-smoothness makes it notoriously hard for auto-differentiation. To circumvent the smoothness issue, we apply the GPU implementation (Wong et al., 2021) of a variance-reduction proximal gradient method SAGA (Defazio et al., 2014). We adopt the *regularization path* approach, in which the solver optimizes over 100 regularization strengths $\lambda_1 > \lambda_2 \cdots > \lambda_{100}$. Here we set $\lambda_1$ to be the strength that returns a model that uses none of the features, and $\lambda_{100} = 0.1 \times \lambda_1$. For LP, we always use $\ell_2$ regularization, we use L-BFGS implemented by scikit-learn, and search for the regularization strength over 100 grids between 0.5 and 6. All the $\lambda$s are evenly spread in the log-space[1]. For FT and SLR-FT-AVD, we select hyperparameters using a training and validation set of size 4 per class. The batch size is fixed to be 512 and the number of epochs is fixed to be 10. We always optimize with AdamW, and choose a cosine rate scheduler with warm-ups. We randomly select learning rate in $[1e-8, 3e-5]$, weight decay in $[0.1, 0.12]$, and warm up steps in $\{0, 50, 500\}$, for 20 trials. The chosen parameters are then fixed throughout all experiments.

**Details about $\boldsymbol{U}, \boldsymbol{W}$**   We include a more detailed description of $\boldsymbol{U}, \boldsymbol{W}$:

$$\boldsymbol{U}_{vd} = \begin{bmatrix} (\boldsymbol{U}_{vd})_1 \\ \vdots \\ (\boldsymbol{U}_{vd})_M \end{bmatrix}, \boldsymbol{U}_{cp} = \begin{bmatrix} (\boldsymbol{U}_{cp})_1 \\ \vdots \\ (\boldsymbol{U}_{cp})_{|\mathcal{C}|} \end{bmatrix}, \boldsymbol{U}_{avd} = \begin{bmatrix} \boldsymbol{U}_{vd} \\ \boldsymbol{U}_{cp} \end{bmatrix}, \text{ where each } \boldsymbol{U}_i \in \mathbb{R}^d.$$

$$\boldsymbol{W}_{vd} = \begin{bmatrix} \mathbf{1} & 0 & \dots & 0 \\ 0 & \mathbf{1} & \dots & 0 \\ \vdots & \vdots & \ddots & \vdots \\ 0 & 0 & \dots & \mathbf{1} \end{bmatrix}, \boldsymbol{W}_{cp} = \begin{bmatrix} 1 & 0 & \dots & 0 \\ 0 & 1 & \dots & 0 \\ \vdots & \vdots & \ddots & \vdots \\ 0 & 0 & \dots & 1 \end{bmatrix}, \boldsymbol{W}_{avd} = \begin{bmatrix} \boldsymbol{W}_{vd} & \boldsymbol{W}_{cp} \end{bmatrix}, \text{where } \mathbf{1} \in \mathbb{R}^{M_c}.$$

Here we assume each class $c$ has the same number of descriptors. The general case can be easily derived. $\boldsymbol{W}_{cp} \in \mathbb{R}^{|\mathcal{C}| \times |\mathcal{C}|}$ is a diagonal matrix. $\boldsymbol{W}_{avd}$ is block-diagonal with $|\mathcal{C}|$ number of rows; each of its block has a row vector of size $M_c$, which amounts to total of $\sum_{c \in \mathcal{C}} M_c = M$ columns.

**What AVDs are selected**   On CIFAR-10, we show that the selected AVDs via sparse logistic regression are intuitive, see table 9.

---

[1]In python numpy.logspace(math.log10($\lambda_1$), math.log10($\lambda_{100}$), 100)

Table 9: Features selected when trained with $\ell_1$ norm on CIFAR-10. The selected important features for each class are intuitive. Notice that the feature selection method does not restrict the candidates to be that particular class's descriptors.

| Classes | Features |
|---|---|
| airplanes | airplanes which has anticollision lights
a photo of airplanes
airplanes which has overhead storage bins |
| cars | cars which has body kit
cars which has bumpers
cars which has wheel arch trim |
| birds | birds which has leg color
birds which has flight silhouette
birds which has eye color |
| cats | cats which has pink tongue
cats which has pink nose
cats which has slit pupils |
| deer | deer which has large facial glands
deer which has long, tufted hair on the neck and shoulders
deer which has short, curved antlers |
| dogs | dogs which has silky fur
dogs which has pattern
dogs which has floppy ears |
| frogs | frogs which has large, bulging eyes
frogs which has ridged or wartylooking skin
frogs which has a fold of skin along the back |
| horses | horses which has hooves
horses which has temperament
horses which has intelligence |
| ships | ships which has lifeboats
ships which has bridge
ships which has bow |
| trucks | trucks which has trailersway control
trucks which has grille
trucks which has lift kits |

## B  More experiment results

As a recap, we use the following acronyms for different methods and datasets. Also see table 2.

**ZS:** Zero-shot classification using text embeddings of hand-crafted prompts ensembles.

**ZS-VD, ZS-AVD:** Zero-shot classification using visual descriptor and augmented visual descriptors, respectively.

**LP:** Linear probing using image embeddings.

**SLR-AVD:** Sparse logistic regression using AVDs.

**FT:** Finetuning the image encoder and classification head.

**SLR-FT-AVD:** Sparse logistic regression with AVD, and then finetune the linear head plus the image encoder with frozen sparsity patterns.

**WISE-FT:** Weight ensemble using ZS and FT.

**WISE-FT+LP:** Weight ensemble using ZS and LP (so only the last linear layer is trained).

**WISE-FT+SLR-AVD:** Weight ensemble using ZS-AVD and SLR-AVD (so only the last linear layer is trained).

**WISE-FT+SLR-FT-AVD:** Weight ensemble using SLR-FT-AVD and ZS-AVD. This is short for WISE-FT+SLR-AVD-FT.

**IN:** ImageNet.

**IN-R:** ImageNet-R.

**IN-A:** ImageNet-A.

**IN-V2:** ImageNetV2.

**IN-Sketch:** ImageNet-Sketch.

The dataset-wise ID-OOD curves of LP vs SLR-AVD on IN-A, IN-R, IN-V2, IN-Sketch, and ObjectNet are listed in figs. 5 to 9, respectively.

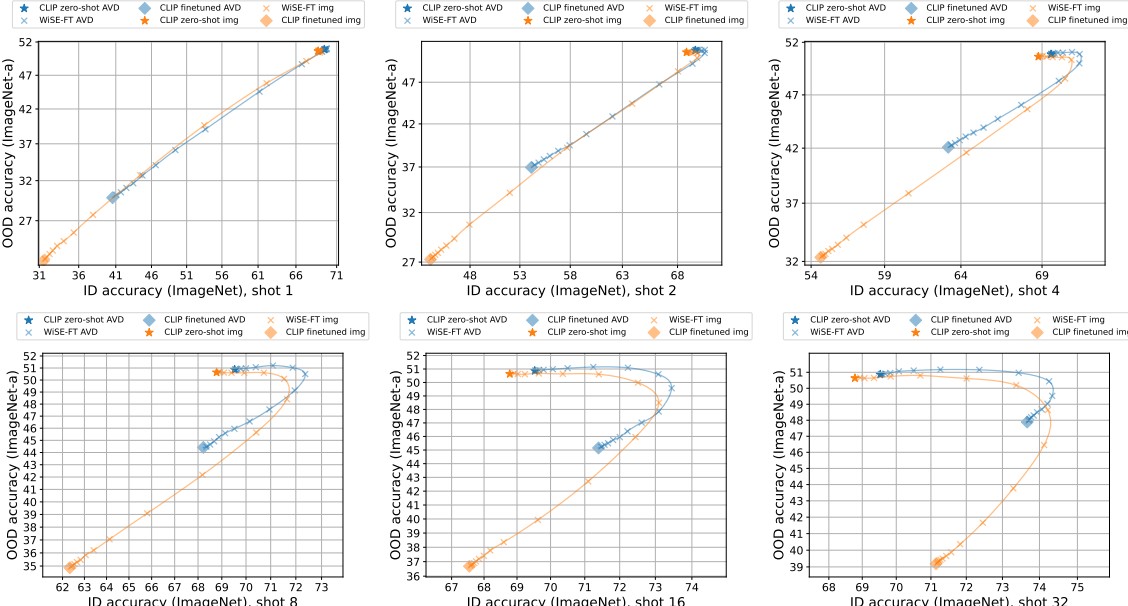

Figure 5: ID-OOD curves of LP vs SLR-AVD on IN-A. $k = 1, 2, 4, 8, 16, 32$.

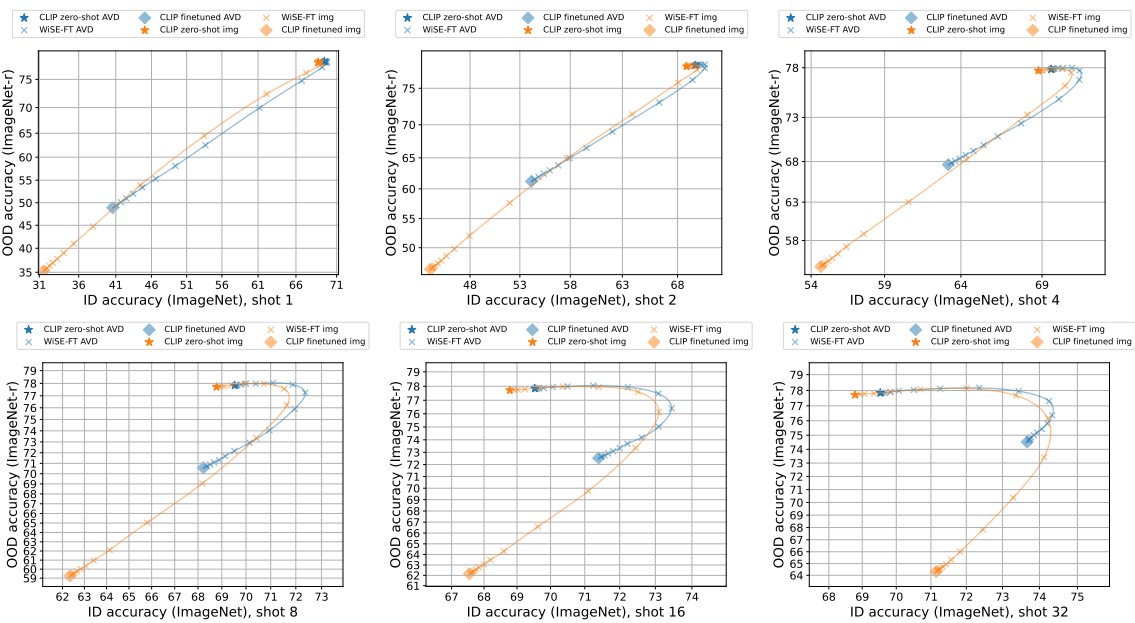

Figure 6: ID-OOD curves of LP vs SLR-AVD on IN-R. $k = 1, 2, 4, 8, 16, 32$.

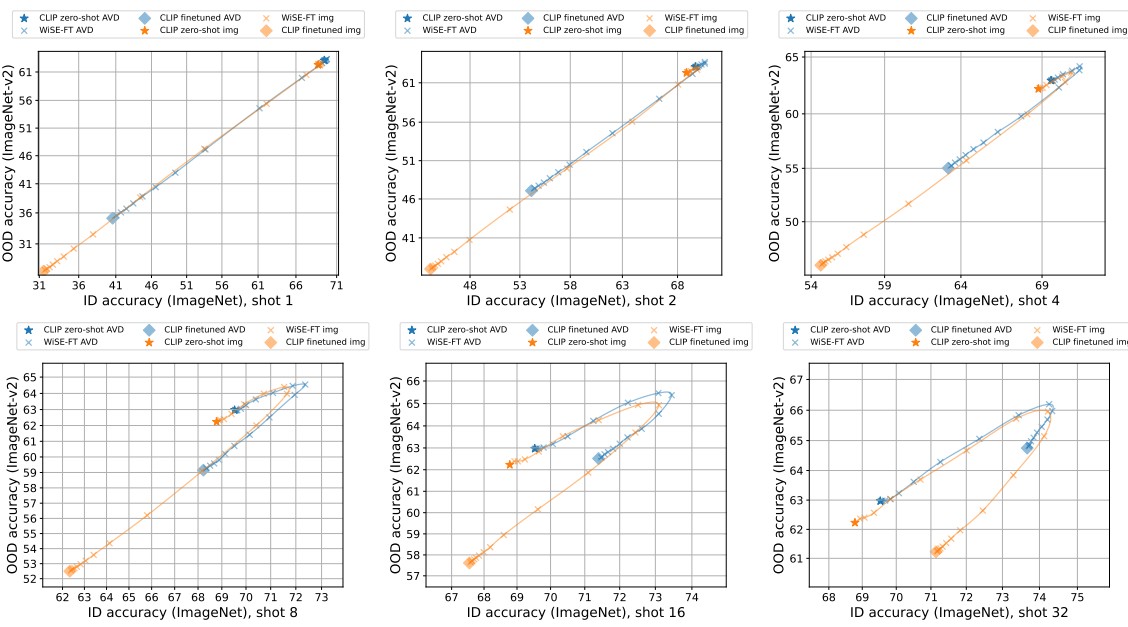

Figure 7: ID-OOD curves of LP vs SLR-AVD on IN-V2. $k = 1, 2, 4, 8, 16, 32$.

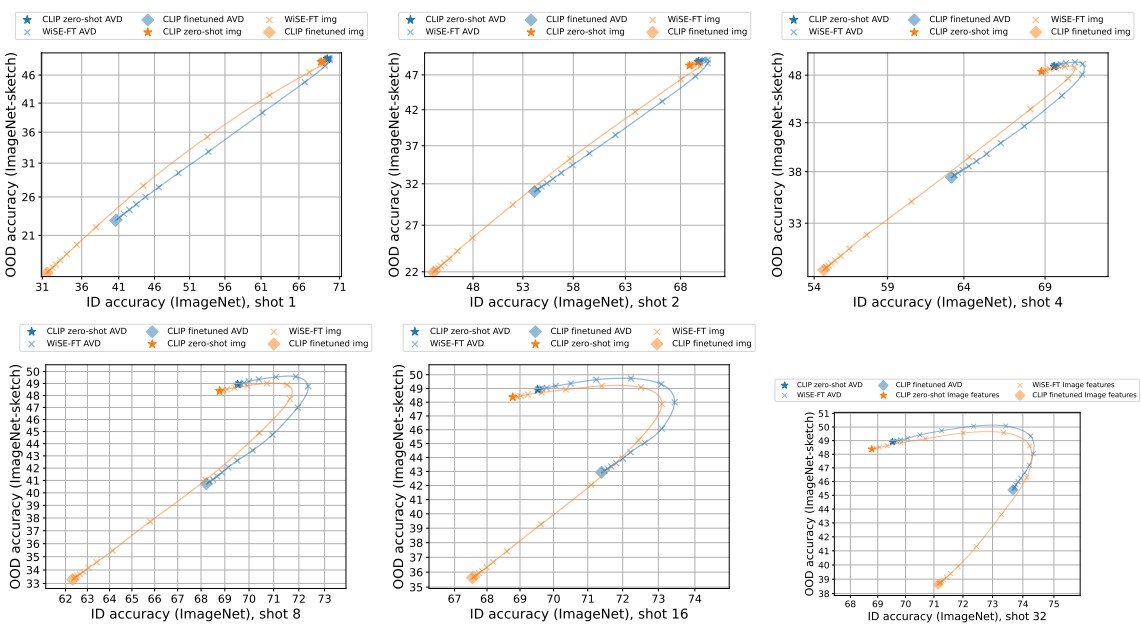

Figure 8: ID-OOD curves of LP vs SLR-AVD on IN-Sketch. $k = 1, 2, 4, 8, 16, 32$.

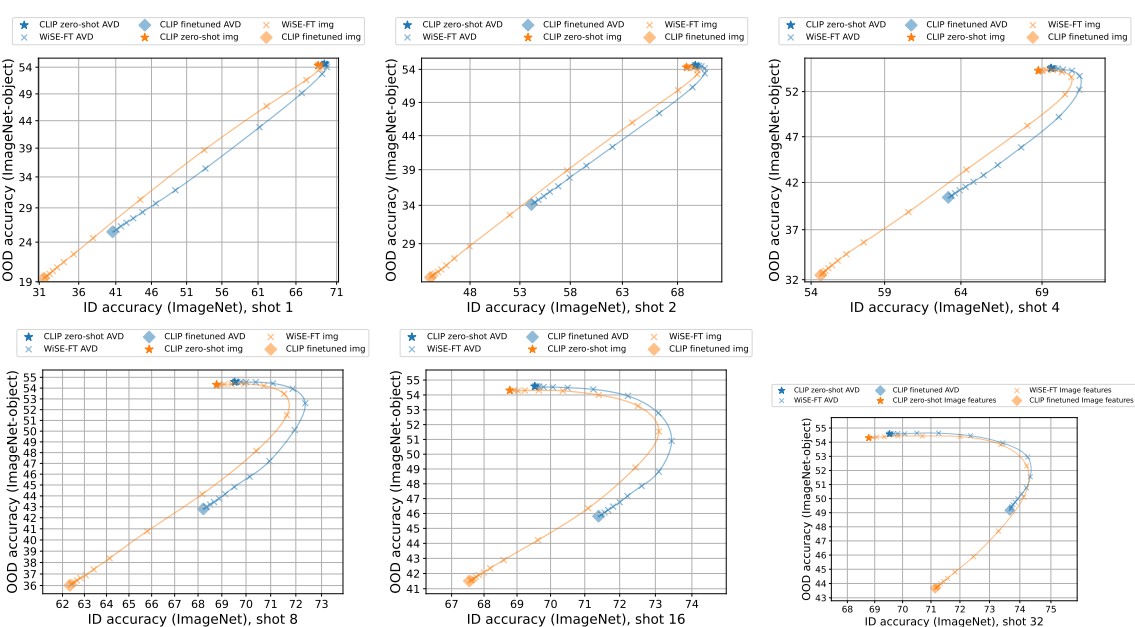

Figure 9: ID-OOD curves of LP vs SLR-AVD on ObjectNet. $k = 1, 2, 4, 8, 16, 32$.

The dataset-wise ID-OOD curves of WISE-FT vs WISE-FT+SLR-FT-AVD on IN-A, IN-R, IN-V2, IN-Sketch, and ObjectNet are listed in figs. 10 to 14, respectively.

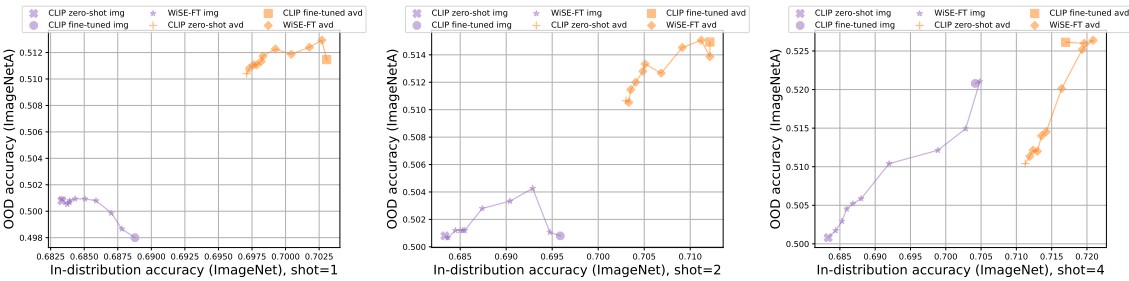

Figure 10: ID-OOD curves of WISE-FT vs WISE-FT+SLR-FT-AVD on IN-A. $k = 1, 2, 4$.

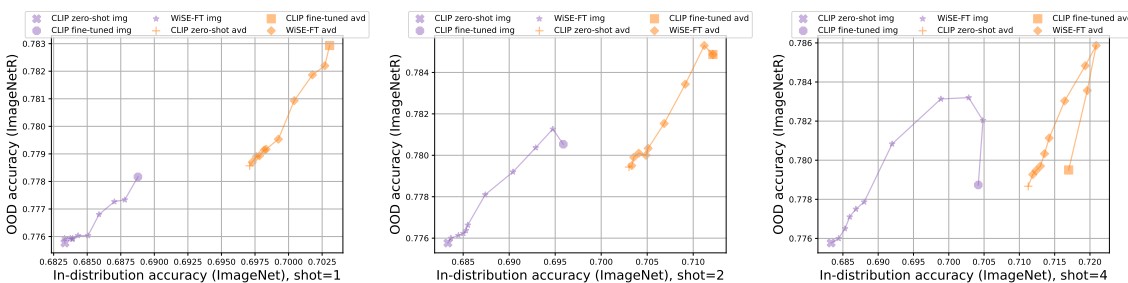

Figure 11: ID-OOD curves of WISE-FT vs WISE-FT+SLR-FT-AVD on IN-R. $k = 1, 2, 4$.

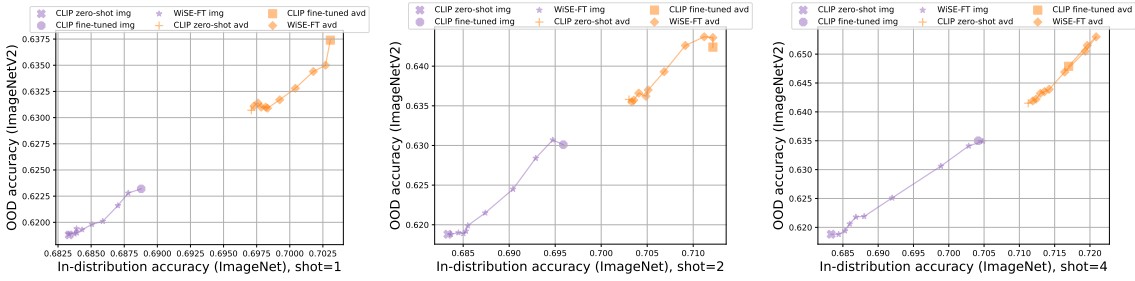

Figure 12: ID-OOD curves of WISE-FT vs WISE-FT+SLR-FT-AVD on IN-V2. $k = 1, 2, 4$.

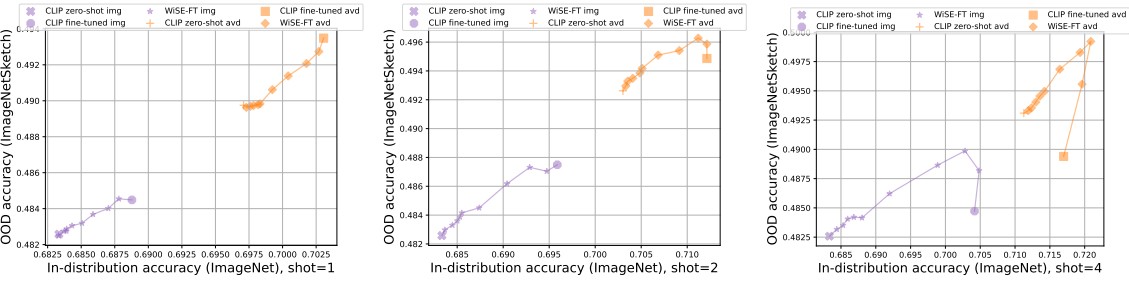

Figure 13: ID-OOD curves of WISE-FT vs WISE-FT+SLR-FT-AVD on IN-Sketch. $k = 1, 2, 4$.

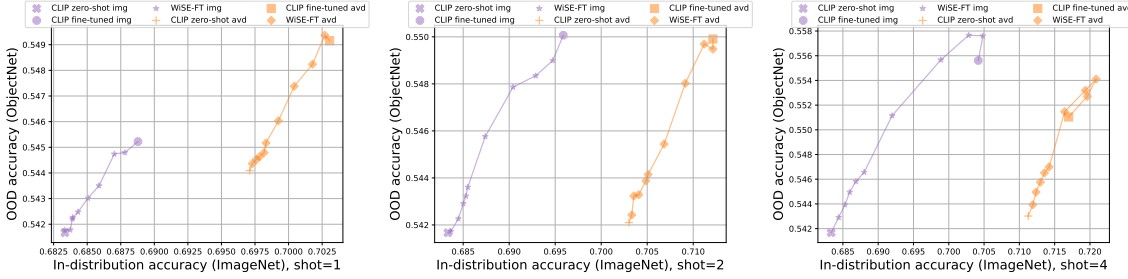

Figure 14: ID-OOD curves of WISE-FT vs WISE-FT+SLR-FT-AVD on ObjectNet. $k = 1, 2, 4$.

The detailed accuracies of WISE-FT vs WISE-FT+SLR-FT-AVD with different choices of $\alpha$ are given in table 13 (ID) and table 12 (OOD). $\alpha = 0$ corresponds to zero-shot accuracy, and $\alpha = 1$ corresponds to full fine-tuned model. The results in the same setting with only the last linear layer trained are presented in table 10 (ID) and table 11 (OOD).

Table 10: Accuracies on ImageNet with difference choices of $\alpha$. We compare LP vs SLR-AVD.

| Shots | $k = 1$ | | $k = 2$ | | $k = 4$ | | $k = 8$ | | $k = 16$ | | $k = 32$ | |
|---|---|---|---|---|---|---|---|---|---|---|---|---|
| Methods $\alpha$ | LP | AVD | LP | AVD | LP | AVD | LP | AVD | LP | AVD | LP | AVD |
| 0.0000 | 68.78 | 69.53 | 68.78 | 69.53 | 68.78 | 69.53 | 68.78 | 69.53 | 68.78 | 69.53 | 68.78 | 69.53 |
| 0.0001 | 68.79 | 69.54 | 68.83 | 69.54 | 68.85 | 69.54 | 68.87 | 69.55 | 68.92 | 69.55 | 68.94 | 69.55 |
| 0.0002 | 68.81 | 69.54 | 68.88 | 69.55 | 68.91 | 69.55 | 68.93 | 69.56 | 69.02 | 69.56 | 69.07 | 69.56 |
| 0.0004 | 68.86 | 69.56 | 68.96 | 69.58 | 69.03 | 69.58 | 69.09 | 69.59 | 69.24 | 69.59 | 69.35 | 69.59 |
| 0.0008 | 68.94 | 69.59 | 69.14 | 69.63 | 69.26 | 69.64 | 69.40 | 69.64 | 69.65 | 69.66 | 69.84 | 69.68 |
| 0.0016 | 69.06 | 69.62 | 69.38 | 69.72 | 69.63 | 69.74 | 69.93 | 69.79 | 70.36 | 69.79 | 70.70 | 69.83 |
| 0.0032 | 69.22 | 69.72 | 69.73 | 69.86 | 70.17 | 69.93 | 70.73 | 70.00 | 71.40 | 70.07 | 72.01 | 70.08 |
| 0.0063 | 68.99 | 69.81 | 69.69 | 70.06 | 70.71 | 70.22 | 71.54 | 70.40 | 72.52 | 70.50 | 73.38 | 70.51 |
| 0.0126 | 67.31 | 69.83 | 68.04 | 70.33 | 70.35 | 70.75 | 71.65 | 71.09 | 73.10 | 71.25 | 74.22 | 71.27 |
| 0.0251 | 62.15 | 69.26 | 63.91 | 70.33 | 68.12 | 71.18 | 70.42 | 71.88 | 72.45 | 72.23 | 74.12 | 72.36 |
| 0.0501 | 53.51 | 66.75 | 57.69 | 69.30 | 64.35 | 71.17 | 68.18 | 72.37 | 71.10 | 73.08 | 73.30 | 73.44 |
| 0.1000 | 44.41 | 61.19 | 51.99 | 66.38 | 60.59 | 69.99 | 65.81 | 71.96 | 69.62 | 73.45 | 72.46 | 74.26 |
| 0.2000 | 37.91 | 53.68 | 47.94 | 62.06 | 57.59 | 67.75 | 64.13 | 70.95 | 68.60 | 73.09 | 71.83 | 74.34 |
| 0.3000 | 35.35 | 49.40 | 46.43 | 59.57 | 56.41 | 66.32 | 63.42 | 70.15 | 68.19 | 72.62 | 71.58 | 74.21 |
| 0.4000 | 34.05 | 46.60 | 45.63 | 57.93 | 55.83 | 65.43 | 63.06 | 69.52 | 67.99 | 72.22 | 71.44 | 74.06 |
| 0.5000 | 33.22 | 44.72 | 45.12 | 56.82 | 55.44 | 64.80 | 62.82 | 69.14 | 67.85 | 72.00 | 71.33 | 73.94 |
| 0.6000 | 32.67 | 43.44 | 44.79 | 55.98 | 55.18 | 64.30 | 62.66 | 68.89 | 67.75 | 71.81 | 71.27 | 73.85 |
| 0.7000 | 32.27 | 42.47 | 44.52 | 55.39 | 54.99 | 63.92 | 62.54 | 68.68 | 67.70 | 71.67 | 71.23 | 73.77 |
| 0.8000 | 31.96 | 41.71 | 44.31 | 54.87 | 54.85 | 63.62 | 62.45 | 68.51 | 67.62 | 71.57 | 71.20 | 73.73 |
| 0.9000 | 31.69 | 41.11 | 44.17 | 54.47 | 54.74 | 63.39 | 62.39 | 68.36 | 67.59 | 71.48 | 71.17 | 73.71 |
| 1.0000 | 31.51 | 40.56 | 44.06 | 54.16 | 54.66 | 63.19 | 62.33 | 68.23 | 67.55 | 71.40 | 71.15 | 73.67 |

Table 11: Accuracies on ImageNet variations with difference choices of $\alpha$. We compare LP vs SLR-AVD. The results are averaged over all 5 ImageNet variations.

| Shots | $k = 1$ | | $k = 2$ | | $k = 4$ | | $k = 8$ | | $k = 16$ | | $k = 32$ | |
|---|---|---|---|---|---|---|---|---|---|---|---|---|
| $\alpha$ ⟍ Methods | LP | AVD | LP | AVD | LP | AVD | LP | AVD | LP | AVD | LP | AVD |
| 0.0000 | 58.66 | 59.03 | 58.66 | 59.03 | 58.66 | 59.03 | 58.66 | 59.03 | 58.66 | 59.03 | 58.66 | 59.03 |
| 0.0001 | 58.67 | 59.03 | 58.68 | 59.03 | 58.69 | 59.04 | 58.70 | 59.04 | 58.70 | 59.04 | 58.71 | 59.04 |
| 0.0002 | 58.68 | 59.02 | 58.68 | 59.04 | 58.70 | 59.04 | 58.71 | 59.04 | 58.70 | 59.04 | 58.73 | 59.04 |
| 0.0004 | 58.69 | 59.03 | 58.70 | 59.04 | 58.71 | 59.04 | 58.73 | 59.03 | 58.75 | 59.05 | 58.81 | 59.05 |
| 0.0008 | 58.70 | 59.03 | 58.75 | 59.06 | 58.81 | 59.06 | 58.84 | 59.06 | 58.88 | 59.06 | 58.99 | 59.06 |
| 0.0016 | 58.72 | 59.06 | 58.81 | 59.08 | 58.90 | 59.09 | 59.03 | 59.11 | 59.07 | 59.08 | 59.23 | 59.11 |
| 0.0032 | 58.69 | 59.07 | 58.77 | 59.13 | 58.97 | 59.18 | 59.15 | 59.19 | 59.20 | 59.16 | 59.47 | 59.21 |
| 0.0063 | 58.30 | 59.10 | 58.27 | 59.14 | 58.79 | 59.23 | 58.89 | 59.31 | 58.98 | 59.28 | 59.40 | 59.37 |
| 0.0126 | 56.77 | 58.90 | 56.41 | 59.16 | 57.40 | 59.30 | 57.56 | 59.45 | 57.79 | 59.49 | 58.34 | 59.59 |
| 0.0251 | 52.55 | 58.09 | 51.93 | 58.70 | 54.30 | 59.14 | 54.81 | 59.38 | 55.47 | 59.55 | 56.28 | 59.78 |
| 0.0501 | 45.05 | 55.46 | 45.65 | 57.11 | 49.68 | 58.20 | 51.10 | 58.73 | 52.54 | 59.13 | 53.86 | 59.76 |
| 0.1000 | 36.63 | 50.25 | 39.71 | 53.88 | 45.30 | 56.12 | 47.77 | 57.21 | 50.03 | 58.05 | 51.87 | 59.25 |
| 0.2000 | 30.30 | 43.39 | 35.56 | 49.45 | 42.08 | 53.33 | 45.47 | 55.21 | 48.38 | 56.46 | 50.61 | 58.29 |
| 0.3000 | 27.83 | 39.70 | 33.94 | 47.00 | 40.79 | 51.76 | 44.55 | 54.02 | 47.73 | 55.60 | 50.12 | 57.70 |
| 0.4000 | 26.59 | 37.37 | 33.09 | 45.44 | 40.10 | 50.76 | 44.09 | 53.25 | 47.39 | 55.02 | 49.88 | 57.31 |
| 0.5000 | 25.82 | 35.84 | 32.57 | 44.43 | 39.67 | 50.12 | 43.79 | 52.74 | 47.18 | 54.63 | 49.71 | 57.04 |
| 0.6000 | 25.28 | 34.73 | 32.22 | 43.70 | 39.40 | 49.63 | 43.59 | 52.36 | 47.03 | 54.37 | 49.62 | 56.82 |
| 0.7000 | 24.87 | 33.94 | 31.99 | 43.16 | 39.18 | 49.25 | 43.45 | 52.07 | 46.92 | 54.16 | 49.55 | 56.67 |
| 0.8000 | 24.59 | 33.33 | 31.80 | 42.72 | 39.03 | 48.95 | 43.34 | 51.86 | 46.83 | 54.02 | 49.49 | 56.54 |
| 0.9000 | 24.36 | 32.85 | 31.65 | 42.38 | 38.91 | 48.71 | 43.26 | 51.67 | 46.77 | 53.88 | 49.44 | 56.45 |
| 1.0000 | 24.17 | 32.42 | 31.53 | 42.08 | 38.80 | 48.51 | 43.19 | 51.54 | 46.72 | 53.79 | 49.41 | 56.35 |

Table 12: Accuracies on ImageNet variations with difference choice of $\alpha$. We compare WISE-FT combined with FT and SLR-FT-AVD, respecively. The results are averaged over 5 ImageNet variations.

| Shot | $k = 1$ | | $k = 2$ | | $k = 4$ | |
|---|---|---|---|---|---|---|
| $\alpha$ ⟍ Method | FT | SLR-FT-AVD | FT | SLR-FT-AVD | FT | SLR-FT-AVD |
| 0.00 | 58.39 | 59.07 | 58.39 | 59.21 | 58.39 | 59.33 |
| 0.02 | 58.40 | 59.09 | 58.40 | 59.22 | 58.45 | 59.39 |
| 0.04 | 58.40 | 59.11 | 58.44 | 59.27 | 58.53 | 59.45 |
| 0.06 | 58.42 | 59.11 | 58.46 | 59.31 | 58.62 | 59.49 |
| 0.08 | 58.42 | 59.12 | 58.48 | 59.33 | 58.69 | 59.58 |
| 0.10 | 58.44 | 59.14 | 58.51 | 59.38 | 58.73 | 59.63 |
| 0.20 | 58.46 | 59.20 | 58.65 | 59.48 | 59.07 | 59.97 |
| 0.40 | 58.50 | 59.29 | 58.82 | 59.68 | 59.40 | 60.24 |
| 0.60 | 58.55 | 59.38 | 58.97 | 59.80 | 59.60 | 60.37 |
| 0.80 | 58.56 | 59.44 | 58.98 | 59.75 | 59.68 | 60.19 |
| 1.00 | 58.58 | 59.49 | 58.98 | 59.74 | 59.50 | 59.88 |

Table 13: Accuracies on ImageNet with difference choice of $\alpha$. We compare WISE-FT combined with FT and SLR-FT-AVD, respecively.

| Shot | $k = 1$ | | $k = 2$ | | $k = 4$ | |
|---|---|---|---|---|---|---|
| Method $\alpha$ | FT | SLR-FT-AVD | FT | SLR-FT-AVD | FT | SLR-FT-AVD |
| 0.00 | 68.33 | 69.71 | 68.33 | 70.30 | 68.33 | 71.13 |
| 0.02 | 68.33 | 69.73 | 68.37 | 70.33 | 68.44 | 71.19 |
| 0.04 | 68.37 | 69.76 | 68.45 | 70.35 | 68.53 | 71.24 |
| 0.06 | 68.39 | 69.78 | 68.50 | 70.41 | 68.60 | 71.30 |
| 0.08 | 68.39 | 69.82 | 68.53 | 70.49 | 68.68 | 71.36 |
| 0.10 | 68.43 | 69.83 | 68.55 | 70.51 | 68.80 | 71.42 |
| 0.20 | 68.51 | 69.92 | 68.74 | 70.68 | 69.20 | 71.64 |
| 0.40 | 68.59 | 70.04 | 69.04 | 70.91 | 69.89 | 71.93 |
| 0.60 | 68.70 | 70.18 | 69.29 | 71.12 | 70.28 | 72.09 |
| 0.80 | 68.78 | 70.27 | 69.47 | 71.21 | 70.48 | 71.96 |
| 1.00 | 68.88 | 70.31 | 69.59 | 71.21 | 70.42 | 71.70 |

**Choosing $\gamma$ and LLM prompting**  We consider another prompt "Give me 100 useful visual features for distinguishing {} in a photo", and use it with frequency penalty (FP) 0 in ①, FP 0.1 in ②. ③ uses the GPT3 prompts in the main text with 0 FP. Unless otherwise specified, other experiments use $\gamma = \frac{1}{M_c+1}$ and FP 0.1, and the GPT prompts in the main text. We find that the GPT3 prompt itself does not matter as much as FP – it is more important to generate a more diverse set of VD. Note in the main text we set $\gamma = 5$, this is because on ImageNet it is hard to guarantee the same $M_c$ across classes (due to an excess number of classes), hence we use a large $\gamma$ to enforce ZS-AVD relies mostly on the strong class prompts. In this ablation study, we enforce GPT to give 100 VDs per class so we can simply average over them.

Table 14: ZS ablation on $\gamma$ and GPT prompts.

| $\gamma$ or prompts | $1/(M_c + 1)$ | 1 | 5 | ① | ② | ③ |
|---|---|---|---|---|---|---|
| CIFAR10 | 91.51 | 91.19 | 91.16 | 91.25 | 91.42 | 90.44 |
| CIFAR10.1 | 86.35 | 85.90 | 85.90 | 85.90 | 85.60 | 85.40 |
| CIFAR10.2 | 83.80 | 83.10 | 83.10 | 83.20 | 84.20 | 82.50 |

Here we include a comparison among SLR-AVD+WISE-FT, CoOp+WISE-FT, WaffleCLIP+WISE-FT, and sparse class prompts (SCP)+WISE-FT. For SCP, we have 1000 classes and 7 templates. We create a text embedding for each class with each template, and get in total 7000 class prompts. On the few-shot training data, we learn a sparse combination of these 7000 features with an $\ell_1$-regularized cross-entropy loss. Finally, we perform WISE-FT with the zeroshot weights.

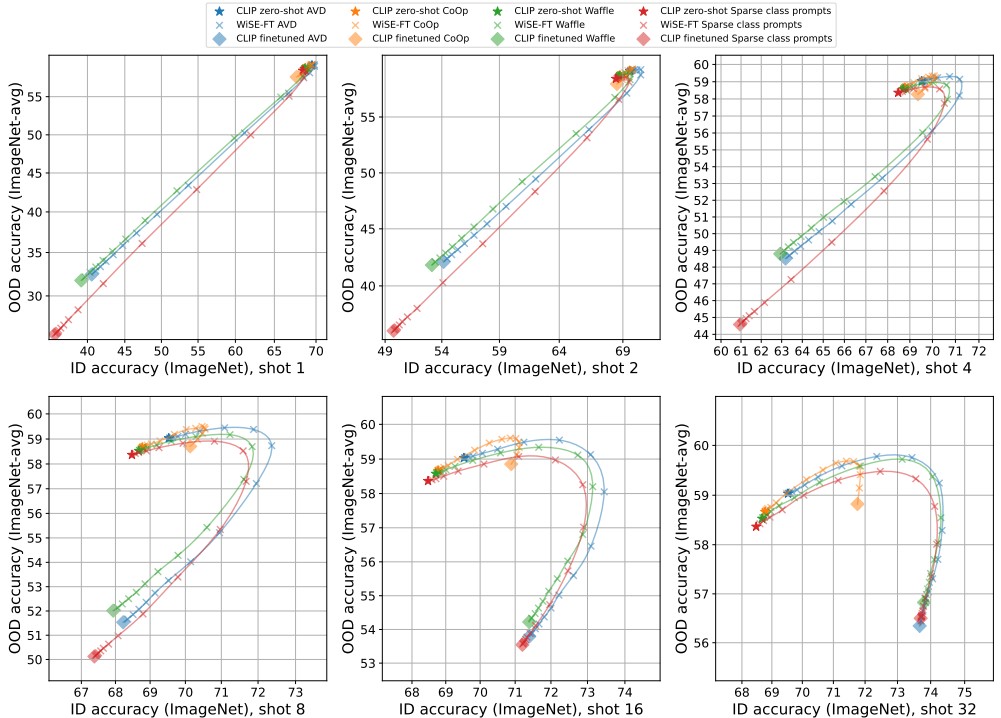

Figure 15: ID-OOD curves of SLR-AVD vs CoOp, Waffle, and the class prompts on average over ImageNet variations. $k = 1, 2, 4, 8, 16, 32$.

**Performance of SLR with AVD vs image features when $k$ is large.** The numbers are averaged over two random runs.

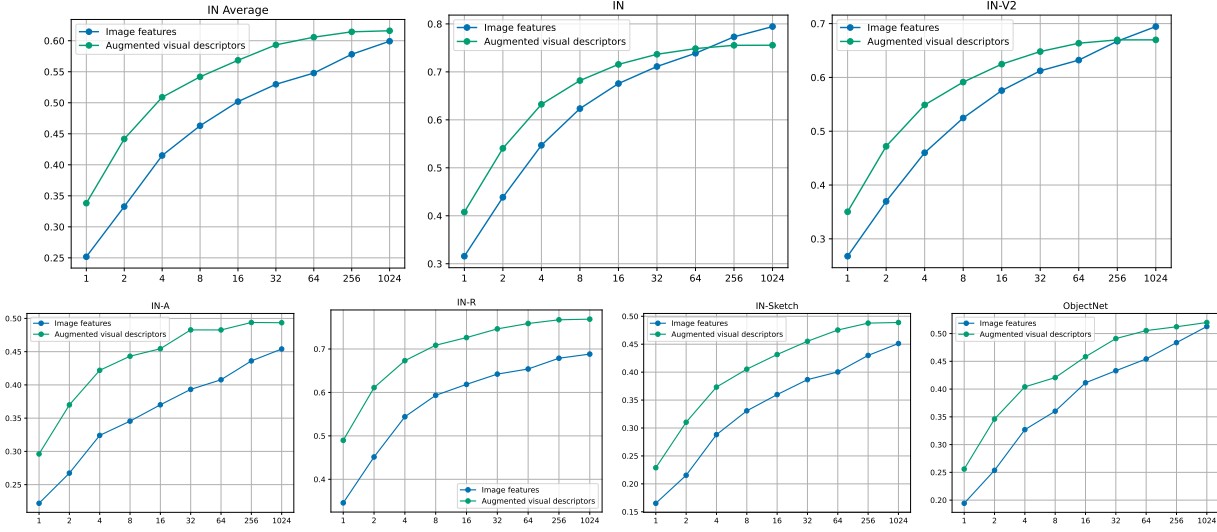

Figure 16: LP vs SLR-AVD, across various ImageNet variations. $k$=1,2,4,8,16,32,64,256,1024.

**How sparse is SLR-AVD** The average number of non-zero entries for each class is $447, 248, 182, 177, 173$, and $135$ for $k = 1, 2, 4, 8, 16, 32$. The numbers are rounded to the nearest integers.

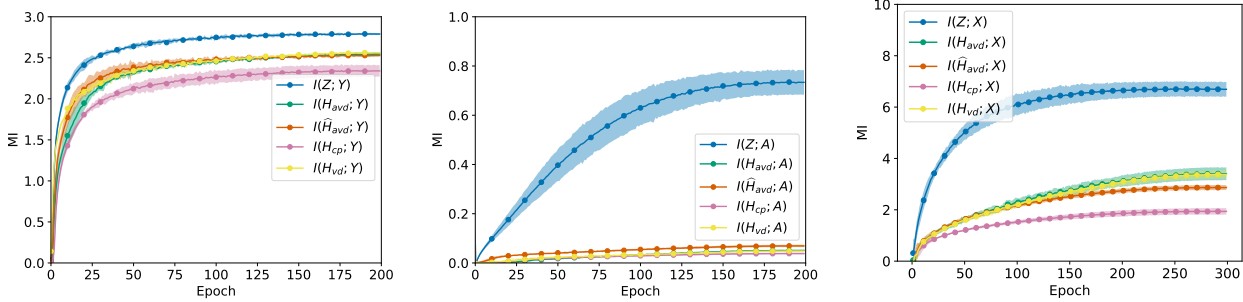

Figure 17: The MI estimations at interest. The estimator is variational and we include the whole optimization trajectory. Each experiment is run three times and two standard deviations are filled. **Left:** the MI between a different set of features and the labels. **Middle:** the MI between a different set of features and the domain indices. **Right**: the MI between a different set of features and the input images.

**Experiments aggregation with** $k = 4$   We include the results with $k = 4$ for our method and several baselines in table 15. The MLP model has layers $512, 4500, 1000$. This amounts to total of $512 * 4511 + 4500 * 1000 = 6804000$ parameters, which equals to the total number of parameters in our SLR model (it has $6804 * 1000$ parameters if we also count the 0 entries).

Table 15: An aggregation of several methods.

|  | IN | IN-V2 | IN-R | IN-A | IN-Sketch | ObjectNet |
|---|---|---|---|---|---|---|
| ZS | 68.78 | 62.23 | 77.72 | 50.64 | 48.38 | 54.31 |
| ZS-VD | 65.89 | 59.19 | 72.75 | 46.11 | 44.84 | 49.60 |
| ZS-AVD | 69.52 | 62.97 | 77.85 | 50.87 | 48.91 | 54.58 |
| MLP weight decay 0.01 | 52.55 | 49.23 | 29.08 | 44.39 | 27.15 | 29.71 |
| MLP weight decay 0.1 | 52.62 | 49.33 | 29.07 | 44.39 | 27.2 | 29.79 |
| LP | 54.66 | 45.92 | 54.5 | 32.34 | 28.77 | 32.44 |
| WISE-FT+LP | 70.71 | 63.66 | 77.89 | 50.69 | 48.89 | 54.39 |
| SLR-AVD | 63.19 | 55.02 | 67.64 | 42.09 | 37.43 | 40.39 |
| WISE-FT+SLR-AVD | 71.18 | 64.2 | 77.98 | 51.07 | 49.37 | 54.59 |
| FT | 70.42 | 63.50 | 77.87 | 52.08 | 48.47 | 55.56 |
| WISE-FT+FT | 70.48 | 63.5 | 78.32 | 52.11 | 48.99 | 55.77 |
| SLR-FT-AVD | 71.70 | 64.79 | 77.95 | 52.61 | 48.94 | 55.10 |
| WISE-FT+SLR-FT-AVD | 72.09 | 65.3 | 78.59 | 52.64 | 49.92 | 55.41 |
| CoOp | 69.36 | 62.77 | 76.54 | 50.43 | 47.96 | 53.68 |
| WISE-FT+CoOp | 70.14 | 63.48 | 78.1 | 51.48 | 49.16 | 54.79 |

