# OpenReview forum: "Text Descriptions are Compressive and Invariant Representations for Visual Learning"
_TMLR — Accepted by TMLR_

### Review · Reviewer_om1n · 2024-03-24

**Summary Of Contributions:**

The paper proposes SLR-AVD, a few-shot method in a VLM classification task. The proposed method first generates multiple descriptions for each class using LLM, and selects important descriptions using sparse logistic regression optimization on descriptive features. Using sparse features is more robust to domain shifts than traditional image embeddings. The authors also show the robustness from an information-theoretic perspective. The authors compare the proposed method in the zero-shot and few-shot settings to demonstrate the effectiveness of their method.

**Audience:**

Yes

**Claims And Evidence:**

No

**Requested Changes:**

The proposals are mainly described in the Weaknesses. I request the authors to change (or add) the explanations/experiments/plots mentioned above.

**minor**
- Missing definition of notation blkdiag in Section 3.1.
- Missing definitions of notation M, P, and d in Section 2.1.
- In the second paragraph in Section 4.2, it might be Figure 3, not Table 1?
- I think the last paragraph in Section 4.2 is more related to Section 4.3?

**Strengths And Weaknesses:**

**Strengths**
- The paper investigates text descriptive features from an information-theoretic perspective, and uses selected descriptive features for a classification task in VLM, which is a new approach.
- Theoretical understanding of text descriptive features is important for reliability.
- Extensive studies on a few-shot setting strengthen the proposed method.

**Weaknesses**
- Ablation study on W^hat_avd : The proposed SLR-AVD method finetunes the entries in W^hat_avd. What are the reasons behind finetuning them? I thought this wouldn't be needed if the features were properly selected by sparse logistic regression. In that sense, 1) what is the performance when not W^hat_avd but only the image encoder is finetuned, and 2) when W_avd is used (and finetuned) instead of W^hat_avd?
- Lack of information-theoretic understanding of sparsity on W : In Section 3.3, the authors explained the benefits of text descriptive features in the information-theoretic perspective. While the paper's proposed method, SLR-AVD, uses W^hat_avd (i.e. sparsity on W), it has not been theoretically compared to W.
- How does the plot for I(H_vd;Y), I(H_vd; A), and I(H_vd; X) look like in Figure 2?
- I suggest to standardize the term when describing the author's main method. For instance, 1) in Figure 3, I assume "augmented visual descriptors" might indicate SLR-AVD, and 2) in Table 1, AVD might indicate SLR-AVD, and 3) In Table 5, AVD should be WISE-SLR as described in page 7 (please correct me if I'm wrong). Also, method definitions in Table 2 should be explained in Section 4.
- In Table 1, to highlight the method of using sparsity, AVD could be compared to SLR-AVD.
- What is the contribution in a zero-shot setting? I assume SLR-AVD can only be applied in a few-shot setting. In that sense, the zero-shot setting only applies to VD and AVD. Since VD is the same as [1] and AVD is simply adding a standard prompt with VD, I wonder what the contribution is in the zero-shot setting. If there is a limited contribution in this setting, the experimental comparison in Section 4.1 could be removed or shortened.
- Inconsistency of the zero-shot results in Table 1 and references [1,2] : In [1] (correspondingly VD in Table 1) and [2], the results are in par with ZS, while in Table 1 their results are lower much lower than ZS.
- What are the plots for WISE-FT + AVD in Figure 4? Here AVD indicates finetuning both image encoder and W (not W^hat_avd).

[1] S. Menon and C. Vondrick, Visual classification via description from large language models.

[2] K. Roth et al., Waffling around for performance: Visual classification with random words and broad concepts.

---

> ### Author Response · Authors · 2024-04-11
> **Response**
>
> Thank you for your thoughtful review and suggestions! Please find our responses below.
>
> - ‘W^hat_avd ‘: The reason behind the way we finetune them is based on the importance of sparse regularization. We added a new ablation section (as well as the original figure 3, SLR-AVD vs L2 AVD) that shows using l1 greatly outperforms l2. This is why we don’t use $W_{avd}$. The reason that we finetune $\widehat W_{avd}$+image encoder rather than just the image encoder is from [1], which gives a theoretical justification for doing updates on all parameters. Although one can argue that in experiments, even these “justified” methods can have unexpected behavior, and there are many different ways to finetune the whole model, for example, [2]. We want to remark that these adjustments are not crucial within the scope of our paper. The key takeaway is that we can find information-theoretically robust features to domain shift. The full-parameter finetuning is a ‘bonus’ to this finding.
> - 'comparison between SLR-AVD and AVD': it has been shown in figure 3.
> - We have added a new result on the sparsely selected AVD: their MI with y is almost as much as the original set of AVD, plus their MI with x is much smaller. These findings justified the usage of sparsity.
> - ‘Contribution of ZS’. We agree the contribution is limited. The main reason is to have a fluent and self-consistent presentation.
> - ‘Inconsistency of ZS result’. The ZS baseline results in [3, 4] did not incorporate the carefully chosen prompts as we did. We tried CLIP-B/32 ZS with our chosen template from OpenAI github, and get 63.77 accuracy, which outperforms WaffleCLIP 63.31 in [4]. Meanwhile, [4] has a more sophisticated random sequence generation – in the experiments they combined both random words and random chars, where we only used random words. We also have different numbers of descriptors.
> - 'What are the plots for WISE-FT + AVD in Figure 4?' The plots in figure 4 are WISE-FT + $\widehat W_{avd}$. As we mentioned above, we have seen an importance of using l1 in the linear case, so we did not include a WISE-FT+AVD comparison due to computation limitations.
>
> We include the MI estimation for VD as well and include it in figure 17 in the appendix. Overall, the estimation shows similar trend to those of AVD.
>
> [1] Fine-Tuning can Distort Pretrained Features and Underperform Out-of-Distribution
>
> [2] Finetune like you pretrain: Improved finetuning of zero-shot vision models
>
> [3] VISUAL CLASSIFICATION VIA DESCRIPTION FROM LARGE LANGUAGE MODELS
>
> [4] Waffling around for Performance: Visual Classification with Random Words and Broad Concepts

---

### Review · Reviewer_3Luv · 2024-03-25

**Summary Of Contributions:**

The paper presents a novel approach to few-shot image classification using text descriptions of images. The proposed method, SLR-AVD, leverages multiple automatically generated visual descriptors for each class to improve classifier performance beyond zero-shot. The main insight is that these features are more compressive and invariant to domain shifts than traditional image embeddings, leading to better generalization in out-of-distribution tasks. The paper provides a thorough analysis of the proposed method and demonstrates its effectiveness through experiments on several image datasets, mainly ImageNet variants.

**Audience:**

Yes

**Broader Impact Concerns:**

A possible concern is about using LLMs to generate descriptors. Such a system would inherit any bias present in the original training data/model, this could potentially lead to biased/unfair classification.

**Claims And Evidence:**

Yes

**Requested Changes:**

1. Robustness to LLM choice: Relying on closed models limits the reproducibility of the work. How much would the performance change if the descriptors are generated by openly-available models, e.g. Lllama2 or Mistral? Showing robustness to the LLM choice would strengthen the work.
2. Comparison with recent prompt-based techniques: Although it is not critical to evaluate the proposed method, the analysis would greatly benefit from a more thorough comparison with prompt-learning techniques like CoCoop [1].

[1] Zhou, Kaiyang, et al. "Conditional prompt learning for vision-language models." Proceedings of the IEEE/CVF conference on computer vision and pattern recognition. 2022.

**Strengths And Weaknesses:**

# Strengths

1. Information-theoretic principles: The paper provides an analysis based on information-theoretic principles, such as compression and invariance, which provide a solid theoretical foundation for the proposed method.
2. Experimental evaluations: The paper provides a comprehensive analysis of the proposed method through experiments on several image datasets, demonstrating its effectiveness compared to baselines.
3. Improved generalization: The proposed method is shown to be robust to domain shifts, as the automatically generated visual features are less informative about the domain, leading to better generalization in out-of-distribution tasks.
4. Interpretability: The suggested approach allows for the identification of the most informative features and their direct textual counterparts, as displayed in Table 6.️

# Weaknesses
1. Lack of ablation studies: A more detailed analysis of the contribution of each component of the proposed method (e.g., language model, ℓ 1 -regularized logistic regression) could help better understand its effectiveness and identify potential areas for improvement.
2. Dependence on high-quality language models: The effectiveness of the proposed method relies on the quality of the language model used to generate the visual descriptors. Any errors or biases in the generated descriptions could impact the overall performance of SLR-AVD. An evaluation with openly-available models would also improve the reproducibility of the paper.
3. Missing comparisons: Although the authors provide a comparison with CoOp in Section 4.4, the analysis would benefit from comparing also with more recent methods, such as CoCoop [1].

---

> ### Author Response · Authors · 2024-04-11
> **Response**
>
> Thank you for your thoughtful review and suggestions! Please find our responses below.
>
> - ‘Ablation’ and language model. We have added a subsection on ablation study. In particular, we show the importance of sparse regularization, and the choice of LLM is not as significant. Figure 3 also had comparisons to both L2 AVD, which shows the importance of l1.
> - We added a comparison to LaBo since that is more relevant to our method (CoCoOp was designed for generalizing to unseen classes). We also included a discussion on CoCoOp in the experiment section.

---

### Review · Reviewer_QRYi · 2024-03-29

**Summary Of Contributions:**

This paper studied using text descriptions as feature exactors for few-shot image classification. The method is very simple. First, LLM (GPT-3) generates a list of textual descriptions for each class, along with the prompts containing the class names, as the information bottleneck. Given an image feature (encoded by CLIP), the dot product of the text features and image features is taken as the new feature representation of that image. Then, a logistic regression layer is trained on top of this feature as the classification head. The experiments show the proposed method has better few-shot ability than the selected baselines.

**Audience:**

No

**Broader Impact Concerns:**

Discuss the potential bias inherited from GPT-3 and CLIP.

**Claims And Evidence:**

No

**Requested Changes:**

* Change the experimental setup and get new results to support the claim of robustness. (see weakness 2 and 3).
* Explain the mutual information results (see weakness 4).
* Improve the presentation of the paper (see weakness 5).

**Strengths And Weaknesses:**

Strength
* Using the mutual information to explain why text descriptive features are better than other representations is interesting.

Weakness
1. **Not many new findings.** This is my main concern of this paper. Almost all the findings in this paper were presented in previous work. The zero-shot results have been shown in Pratt et al. (2022) and Menon & Vondrick (2022). The few-shot results have been demonstrated in Yang et al. (2023). The proposed method is almost the same as these works, and this paper just formalizes the ideas of previous work and replicates the results. The main difference between the proposed method and existing ones is the sparse logistic regression and some fine-tuning, but the author did not explain it well and lacked convincing ablations to verify its effectiveness (see weakness 3). The mutual information analysis is interesting, but the results have some issues (see weakness 4). The robustness claims have severe problems (see weakness 2). Overall, from my point of view, this paper does not provide additional important insights to the community.

2. **Evidence for the robustness claim.** Usually, to prove the robustness of a model, the experiments should train/validate and test in different domains. Both in-domain (ID) and out-of-domain (OOD) performance need to be reported, and a robust model should have good in-domain, not drop too much on OOD (smaller gap), and have a better average domain performance than baselines.

    The few-shot setting is not enough to justify the robustness claim. Robustness means the model is less likely to learn the spurious cues/biases of the in-domain data. However, in few-shot settings, the model does not learn much from the in-domain data; it still mainly leverages the priors (from LLM and CLIP) to make the prediction. The improvement over the linear probe is just because the text features have better priors, as shown in LaBo (Yang et al., 2023), not because the proposed method is more robust.

    The proposed method is better than the linear probe in both in-domain and out-of-domain, but the domain gap doesn't get narrower, which means the proposed method may just be a stronger few-shot classifier, not a more robust classifier.

    To support the robustness claim, the author must show the results using more in-domain data to train the model and needs to show the proposed model has a smaller in-domain, out-of-domain gap. In addition, using other more confounded datasets, such as Waterbirds (Sagawa et al., 2020) and [WILD](https://wilds.stanford.edu/datasets/) benchmark could be a better choice.

3. **The evaluation and results are not solid.** This paper lacks direct comparison with existing work, like LaBo (Yang et al., 2023), in few-shot settings. Most improvements are marginal, especially compared with fine-tuning results (Table 4). Why in Table 4 only choose maximum $k=4$, not $32$, same as the other settings? I hypothesize that fine-tuning in small shots makes it hard for the model to update all parameters. The ablation of the sparse logistic regression is also missing. What if we learn a linear layer instead?

4. **Inconsistent Mutual Information Results.** In the second paragraph of section 3.3, it says, "a good feature representation $Z$ would have high mutual information with the label, $I(Z; Y)$, but low MI with the domain index, $I(Z;A)$." However, in paragraph 5, the results are "$I(Z; A) > I(H_{avd}; A) > I(H_{cp}; A)$" which means class prompts (cp) is more invariant to domain shift than augmented visual descriptors (avd). If this is true, why use avd as the feature?

5. **The presentation of the paper can be improved.** Although the idea of the paper is simple, and I am very familiar with this field, I still find this paper hard to read. The method section introduces too many notions, and the method names in the results section are highly abbreviated, which makes it difficult to parse the results. I always need to go back and forth to understand the numbers. The figures in the paper can be improved. Figure 1 is below the standard of the TMLR publication. Figure 2 is not color-blind friendly, and I cannot map the lines to each label. Figures 3 and 4 are also hard to parse. In summary, this paper needs a lot of effort to improve its readability.

---

> ### Author Response · Authors · 2024-04-11
> **Response**
>
> Thank you for your thoughtful review and suggestions! Please find our responses below.
>
> - ‘Robustness claims’. We have conducted new experiments on the WILDS benchmark, specifically with iWildCam and FMoW. We see that on iWildCam SLR-AVD has significantly outperformed linear probing, while on FMoW they are on par. See section 4.6.
> - ‘Comparison to LaBo’. We have added a comparison to LaBo as well as an ablation study in the revision, see section 4.5 and 4.7. In particular, learning a linear layer instead (for example, with just l2 regularization) can lead to great degradation. Figure 3 also had comparisons to both L2 AVD and learning with just CP, which shows the importance of l1 and the advantage of AVD over CP. We were limited by resources to scale up the full parameter finetuning results, so we chose max k=4 instead of 32.
> - Mutual information. Just having low MI with A isn’t enough for learning. Imagine a feature extractor f that outputs a constant c for all input x. Such f has 0 MI with the domain, making it very invariant. However, this feature extractor is useless. The key is to have high MI with Y and low MI with A, which we show to be the case, in both the MI simulation as well as figure 3.
>
> We have also improved figures 1,2,3,4. We also get rid of some of the redundant acronyms. Most notations at the moment seem necessary to give a clear presentation, if you have suggestions on how to reduce the number of notations, we are happy to incorporate them.

---

### Author Response · Authors · 2024-04-11
**To all reviewers**

We thank every reviewer for their thoughtful comments and suggestions. We have added several new experiments in the revision and here is a summary.

1. We have added a new section that compares our method to LaBo [1], the results show that our method outperforms LaBo and linear probing by a large margin. We should remark that the original LaBo paper selects 50000 total concepts for 1000 ImageNet classes, but the l1 regularization in our setting selected a much smaller set of concepts (meanwhile, we cannot really predefine a number of features to select using l1). See details in section 4.5.
2. We add a new section that does full dataset training using iWildCam and FMoW in the WILDS benchmark. On iWildCam, SLR-AVD has outperformed LP by a large margin, and it performs on par with LP on FMoW. See section 4.6.
3. We add a section 4.7 that does an ablation study on SLR-AVD. In particular, we study how (a) sparsity helps the performance, and (b) how much does the LLM matter in generating relevant AVDs. We observe that sparsity is crucial for robustness to OOD, and the choice of LLM does not matter as much. The language understanding ability is limited by the CLIP text encoder. Although not explicitly written down previously in an ablation study section, the comparison in figure 3 has already presented the importance of using l1 regularization, as SLR-AVD outperforms L2 AVD; the importance of using VD in addition to CP is also presented. The corresponding text (that describes figure 3 in the main text) also discussed “specifically, we see that the incorporation of l1 sparsity is crucial for the improvement of the few-shot performance…”.
4. For the mutual information experiment, we in addition measure the MI between the AVD that are selected by SLR-AVD, and three random variables (domain indices A, labels Y, and the input images X). Compare to the original set of all AVDs, the sparsely selected AVDs have (a) nearly the same MI with the labels Y, (b) a smaller MI with A (this is guaranteed by DPI), and (c) a noticeable drop in MI with the input X. Let us call the whole set of AVDs by $H$, and the sparsely selected set of AVDs as $\widehat H$
    - Notice that (b) is guaranteed by DPI. So $\widehat H$ is more robust to OOD.
    - For (a), DPI tells us that $I(\widehat H; Y)\leq I( H; Y)$, so they are less predictive. However, numerical evaluation suggests that the MI does not drop too much. Therefore, most of the predictive power is preserved by the SLR-AVD process.
    - The significant drop in MI with X makes the $\widehat H$ a better compressor of the input X, therefore, they generalize better.

[1] Language in a Bottle: Language Model Guided Concept Bottlenecks for Interpretable Image Classification

---

### Decision · Action_Editor_AyXM · 2024-05-06

**Recommendation:** Accept with minor revision

**Comment:**

The reviewers appreciate the core components of the article, such as the use of information-theoretic principles to build the approach (QRYi, om1n, 3Luv) and the extensive experimental validation (om1n, 3Luv).

At the same time, the reviewers raised several concerns, mostly regarding the technical contribution (i.e. QRYi) and the experimental validation itself. Specifically, the latter was considered lacking important analyses, such as in- vs out-of-domain gaps (QRYi, om1n), changes w.r.t. the number of shots (QRYi), missing comparisons with more recent baselines (QRYi, om1n, 3Luv), lack of extensive ablation studies (3Luv, om1n), and inconsistencies on the mutual information results (QRYi, om1n).

The authors' reply and the updated version of the manuscript addressed most of the concerns, especially those related to the experimental comparisons, ablations, and the number of shots. The only remaining concern is the discrepancy between the empirical results (Figure 2) and some inequalities of Section 3.3: $I(H_{avd}, Y)\geq I(\hat{H}_{avd}, Y)$,

$I(H_{avd}, A)\geq I(\hat{H}_{avd}, A)$.



Nevertheless, all main concerns have been addressed and the previously mentioned misalignment is mild (i.e. leaning toward equality of the two mutual information). Thus, the AE recommends the acceptance of the work. As a suggested minor revision, the last part of Section 3.3 can be expanded to clarify the reasons behind the misalignment and/or why the differences are negligible.

**Audience:**

Few-shot classification with vision-language models is currently a hot topic in multimodal learning. The article discusses some important design choices to tackle this task (i.e. language descriptors, classifier type) and how they impact the model's robustness. Thus, it is definitely interesting for TMLR's audience, especially for researchers in multimodal learning.

**Claims And Evidence:**

This work tackles few-shot learning with vision language models (VLM) by proposing a new pipeline, SLR-AVD, which consists of three steps: 1) using a large-language model (LLM) to generate class descriptions; 2) processing them, together with the available images, to obtain a set of visual descriptions; 3) applying sparse-logistic regression to perform feature selection.

The core claim of the article is that, in the context of VLMs, it is possible to find a subset of features (i.e. a compressed representation) that leads to better invariance to domain-shift, thus better generalization. This is validated via experiments on in and out-of-domain settings (e.g. Table 1, Table 7), as well as ablation studies (e.g. Table 2), and by varying the amount of data available (e.g. Table 5, Section 4.6). The mutual information inequalities are also tested experimentally (Fig. 2). While some remaining aspects need clarification, the experimental results support the claims well.